# Concepts' Information Bottleneck Models

**Karim Galliamov**[1]    **Syed M Ahsan Kazmi**[2]    **Adil Khan**[3]    **Adín Ramírez Rivera**[4]

[1]University of Amsterdam    [2]University of West of England    [3]University of Hull    [4]University of Oslo
[1]karim.galliamov@student.uva.nl    [2]ahsan.kazmi@uwe.ac.uk    [3]a.m.khan@hull.ac.uk    [4]adinr@uio.no

## Abstract

Concept Bottleneck Models (CBMs) aim to deliver interpretable predictions by routing decisions through a human-understandable concept layer, yet they often suffer reduced accuracy and concept leakage that undermines faithfulness. We introduce an explicit Information Bottleneck regularizer on the concept layer that penalizes $I(X; C)$ while preserving task-relevant information in $I(C; Y)$, encouraging minimal-sufficient concept representations. We derive two practical variants (a variational objective and an entropy-based surrogate) and integrate them into standard CBM training without architectural changes or additional supervision. Evaluated across six CBM families and three benchmarks, the IB-regularized models consistently outperform their vanilla counterparts. Information-plane analyses further corroborate the intended behavior. These results indicate that enforcing a minimal-sufficient concept bottleneck improves both predictive performance and the reliability of concept-level interventions. The proposed regularizer offers a theoretic-grounded, architecture-agnostic path to more faithful and intervenable CBMs, resolving prior evaluation inconsistencies by aligning training protocols and demonstrating robust gains across model families and datasets.

## 1 Introduction

In many real-world settings, models must not only be accurate but also provide explanations that humans can scrutinize and act upon. Among explainability approaches, self-explainable models are particularly compelling because they produce structured, intrinsic rationales and facilitate targeted debugging. Concept bottleneck models (CBMs) (Koh et al., 2020) embody this paradigm by predicting through human-interpretable concepts and enabling test-time interventions on those concepts.

Despite their appeal, CBMs often underperform black-box models and suffer from concept leakage (the encoding of extraneous input information into concept activations) which undermines interpretability and weakens intervention reliability (Mahinpei et al., 2021; Margeloiu et al., 2021). Prior efforts to alleviate these issues typically modify architectures or enrich concept embeddings (Havasi et al., 2022; Kim et al., 2023), which introduces design complexity, can reduce intervenability, and may not generalize across CBM variants.

We propose a simple, architecture-agnostic solution grounded in information theory: impose an Information Bottleneck (IB) regularizer directly on the concept layer (Alemi et al., 2017; Tishby et al., 2000). Our objective enforces minimal-sufficient concepts by compressing $I(X; C)$ while preserving $I(C; Y)$, thereby discouraging spurious signal flow into concepts without sacrificing task-relevant information—as shown in Fig. 1. We derive two practical training objectives, including an entropy-based surrogate that replaces the explicit $I(X; C)$ term with an equivalent formulation in terms of entropies. We empirically justify its simplifying assumption on the concept entropy and show it performs on par with a variational bound. The regularizer is plug-and-play: it requires no architectural changes, side channels, or additional supervision, and applies broadly across CBM families.

For our evaluation, first, we follow an intra-method protocol: for each baseline, we compare it against its proposed regularized variant under identical backbones, data, and training recipes. Second, we substantiate interpretability improvements beyond accuracy: IB-regularized CBMs exhibit substantially lower leakage, stronger and more predictable test-time interventions, and clearer information-plane dynamics consistent with the minimal-sufficiency goal. Third, we show stable performance across

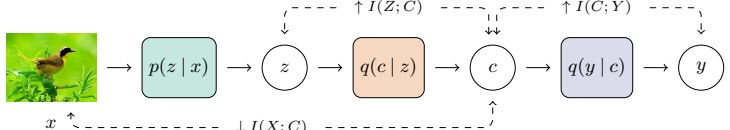 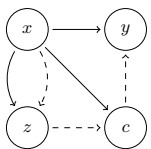

**Figure 1:** Our proposed CIBMs pipeline. The image is encoded through $p(z \mid x)$, which in turn encodes the concepts with $q(c \mid z)$, and the labels are predicted through $q(y \mid c)$. These modules are implemented as neural networks. We introduced the IB regularization as mutual information optimizations over the variables as shown in dashed lines.

**Figure 2:** Our generative model $p(y \mid x)p(c \mid x)p(z \mid x)p(x)$ (solid lines), and its variational approximation $q(y \mid c)q(c \mid z)q(z \mid x)q(x)$ (dashed lines).

a range of IB strengths. These experiments indicate that IB at the concept layer improves both predictive performance and the reliability of concept-level reasoning.

Our contributions are: (i) We establish, to our knowledge, the first direct integration of the Information Bottleneck principle at the concept layer in CBMs, reframing interpretability from a heuristic add-on into a structurally enforced, theoretically grounded property of the model. We provide two concrete, tractable training objectives with empirical support for their assumptions. (ii) We present a comprehensive, reproducible assessment across diverse CBM variants and datasets showing that IB-regularized CBMs consistently improve target accuracy, reduce concept leakage, and enhance intervention effectiveness.

Rather than redesigning CBM architectures or accumulating concept embeddings (Havasi et al., 2022; Kim et al., 2023), our information-theoretic regularization offers a simple, general mechanism to obtain concepts that are both maximally predictive of the target and minimally contaminated by the input—yielding more faithful, intervenable, and performant concept-based models.

## 2 RELATED WORK

### 2.1 CONCEPT BOTTLENECK MODELS

**CBMs.** A CBM (Koh et al., 2020) predicts via $\hat{y} = f(g(x))$, where $x \in \mathbb{R}^D$, $g \colon \mathbb{R}^D \to \mathbb{R}^K$ maps inputs to $K$ human-interpretable concepts, and $f \colon \mathbb{R}^K \to \mathbb{R}$ maps concepts to the target. Training uses triplets $\{(x_i, c_i, y_i)\}_{i=1}^N$ with ground-truth concepts $c_i$, and can proceed independently, sequentially, or jointly (Koh et al., 2020). While CBMs enable reasoning and interventions through concepts, they often trail black-box models in accuracy. Concept Embedding Models (CEM) (Espinosa Zarlenga et al., 2022) improve accuracy by learning "active" and "inactive" embeddings per concept, but require extra regularization for effective test-time interventions and exhibit increasing $I(X; C)$ without compression. In contrast, we retain the native concept space and explicitly regularize it with a concept information bottleneck, incorporating a mutual-information constraint into the loss; this plug-in regularizer is architecture-agnostic and applicable across CBM variants.

**Probabilistic CBMs.** ProbCBMs (Kim et al., 2023) and energy-based CBMs (Xu et al., 2024) model concept uncertainty by predicting distributions over concepts and often rely on anchor points for class mapping. Other works extract concepts without annotations by introducing inductive biases, e.g., via language or label-free supervision (Oikarinen et al., 2023; Yang et al., 2023). We avoid anchor points (since they increase inference cost and introduce additional tuning hyperparameters) and instead learn concepts using a variational approximation to our concept-level information bottleneck.

**Post-hoc CBMs.** Another line of work converts pre-trained models into CBMs. Side-channel and recurrent CBMs (Havasi et al., 2022) introduce, respectively, a parallel concept bottleneck and sequential concept prediction; the former lowers intervenability, while the latter compromises concept disentanglement. Post-hoc CBMs (Yuksekgonul et al., 2023) train concept and class predictors on frozen penultimate-layer embeddings and typically require residual connections for competitive accuracy, leaving residual information paths that concept supervision cannot shape—undermining both interpretability and intervention reliability.

**Concept Leakage.** CBMs are prone to concept leakage (i.e., encoding input information in concept activations beyond intended semantics) across both soft and hard concepts (Mahinpei et al., 2021; Margeloiu et al., 2021). Margeloiu et al. (2021) report that CBM desiderata are best met

under independent training, while joint and sequential training tend to encode extra input information; saliency analyses further suggest that, under all three regimes, concepts are often not grounded in meaningful input features. To quantify leakage, Oracle and Niche Impurity Scores were proposed by Espinosa Zarlenga et al. (2023a). Motivated by these findings, we posit that compressing $I(X; C)$ while preserving $I(C; Y)$ yields more faithful concepts and representations; our experiments across diverse tasks support this hypothesis.

## 2.2 INFORMATION BOTTLENECK

Tishby et al. (2000) introduced the information bottleneck (IB) as the minimization of the functional $\mathcal{L}_{\text{IB}} = I(X; Z) - \beta I(Z; Y)$, where $I(\cdot; \cdot)$ is the mutual information, $\beta$ is the Lagrange multiplier, $X, Y$ and $Z$ are random variables that represents the data, labels, and latent representations, respectively. This functional seeks a representation $Z$ that is maximally predictive of $Y$ while discarding irrelevant information about $X$. This "minimal-sufficient" principle connects to generalization via the memorization-compression dynamics, wherein $I(Z; Y)$ grows throughout training while $I(X; Z)$ first increases (memorization) and later decreases (compression).

Alemi et al. (2017) extended the IB framework to deep neural networks by doing a variational approximation of latent representation. And, Kawaguchi et al. (2023) analyzed the role of IB in estimation of generalization gaps for classification task. Their result implies that by incorporating the IB into learning objective one may get more generalized and robust network. Unlike prior work that applies IB to generic latent features, we target the concept layer in CBMs. To our knowledge, we are the first to formalize and integrate IB at the concept level in CBMs: we introduce the concept variable $C$ and explicitly regularize $I(X; C)$ while preserving $I(C; Y)$, yielding concept representations that are both compact and task-relevant. We derive a tractable upper bound that links predicted concepts and ground-truth signals into a training regularizer that enforces the desired memorization-compression behavior at the concept layer. Practically, this yields objectives realizable via a variational approximation and, alternatively, a mutual-information estimator, enabling seamless integration across common CBM formulations without architectural changes.

## 3 CONCEPTS' INFORMATION BOTTLENECK

Concept Bottleneck Models (CBMs) aim for high interpretability by introducing human-understandable concepts, $C$, as an intermediary between latent representations, $Z$, and the labels $Y$, so that the overall pipeline is $X \to Z \to C \to Y$—cf. Fig. 2. To keep the concepts truly interpretable we must prevent them from encoding irrelevant details of the raw input $X$. This is captured by minimizing the mutual information between the inputs and the concepts, $I(X; C)$, which directly limits the amount of data that can "leak" into $C$. At the same time we want the concepts to be predictive of the label and to be well-supported by the latent representation. Accordingly, we maximize $I(C; Y)$ and $I(Z; C)$. Our initial objective is $\max I(Z; C) + I(C; Y)$, s.t. $I(X; Z) \leq I_C$, where $I_C$ is an information constraint constant, that equivalently is the maximization of the functional of the concepts' information bottleneck (CIB)

$$\mathcal{L}_{\text{CIB}} = I(Z; C) + I(C; Y) - \beta I(X; Z), \tag{1}$$

where $\beta$ is a Lagrangian multiplier. This formulation ensures a strong connection between latents, $Z$, and the concepts, $C$. This means that one wants $Z$ to be maximally useful in shaping the concepts $C$, while also ensuring that the concepts are informative about the target.

While the data processing inequality, $I(X; C) \leq I(X; Z)$, suggests that compressing the latent space $Z$ implicitly limits the information in $C$, we propose enforcing this constraint directly on the concept space. We treat this not merely as a bound, but as a design choice. By explicitly minimizing $I(X; C)$, we strictly control the amount of source information retained in the concepts, regardless of the capacity of $Z$. This prioritizes the cleanliness of the interpretable layer, leading to the objective

$$\mathcal{L}_{\text{CIBM}} = I(Z; C) + I(C; Y) - \beta I(X; C).^1 \tag{2}$$

---

[1]Note that one can obtain the same loss if the optimization problem is constrained over the concepts instead, i.e., $\max I(Z; C) + I(C; Y)$ s.t. $I(X; C) \leq I_C$. Nevertheless, we present the relation with the traditional compression for completeness.

We posit that by compressing the information between the data, $X$, and the concepts, $C$, instead of the latent representations, $Z$, we can control the redundant information of the data within the concepts. By limiting $I(X;C)$, we explicitly control the amount of *redundant or spurious* information that can enter the concepts. In contrast, compressing $X \to Z$ first and then deriving $C$ from $Z$ may leave a pathway for such leakage to survive the $Z \to C$ mapping. Moreover, the bound $\mathcal{L}_{\text{CIBM}}$ (2) can be interpreted as compressing the information between $X$ and $C$ directly. This provides a stronger interpretability constraint than limiting $I(X;Z)$, yielding a more robust compression and, consequently, more faithful concepts.

By shifting the compression from the latent space to the concept space we obtain (i) concepts that are *intrinsically* concise and, thus, easier for a human to interpret, and (ii) a principled safeguard against data leakage that would otherwise contaminate the bottleneck. The CIBM formulation (2) provides a clean information-theoretic framework for building such interpretable CBMs. We propose two implementations of our framework by exploring different ways of solving the mutual information based on a variational approximation of the data distribution.

Our CIBMs provide a tighter generalization bound than traditional CBMs, established through a PAC-Bayes framework (McAllester, 1999; 2003). The derived generalization gap ($\Delta$) reveals the theoretical advantage of the CIBM over the traditional CBM. The result (Theorem 2) demonstrates that the guaranteed True Risk of the CIBM is strictly lower than that of the CBM, provided the regularization strength $\beta$ is sufficiently small to ensure the gap is positive ($\Delta > 0$). This occurs because the CIBM's complexity reduction significantly outweighs the resulting slight increase in empirical training error (i.e., the $\beta$ penalty), leading to a tighter generalization bound. We detail the derivation and analysis of the conditions for this gap in Appendix B.

## 3.1 Bounded CIB

We consider the bound to the concept bottleneck loss (2) in terms of the entropy-based definitions of the mutual information. By using a variational approximation of the data distribution, we bound it as

$$\mathcal{L}_{\text{CIBM}} \geq (1 - \beta) \underset{p(z)}{\mathbb{E}} \left[ H(p(c \,|\, z)) - H\left(p(c \,|\, z), q(c \,|\, z)\right) \right] - \underset{p(c)}{\mathbb{E}} H\left(p(y \,|\, c), q(y \,|\, c)\right), \quad (3)$$

$$= - \mathcal{L}_{\text{S-CIBM}}. \quad (4)$$

We detail this derivation in Appendix A. We can maximize the concepts' information bottleneck by minimizing the cross entropies of the predictive variables, $y$ and $c$, and their corresponding ground truths and by adjusting the entropy of the concepts—cf. Fig. 2. We denote the models that were trained using this loss $\mathcal{L}_{\text{S-CIBM}}$ (4) by $\text{IB}_B$. To implement it, we need to estimate the entropy of the concepts distribution $p(c)$. We give details of this estimator in Appendix D.4.

## 3.2 Estimator-based CIB

Another way to obtain a bound over the concept information bottleneck (2) is to only expand the conditional entropies that are not marginalized (A.1) to avoid widening the gap in the bound, i.e.,

$$\mathcal{L}_{\text{CIBM}} \geq H(Y) + H(C) - \underset{p(c)}{\mathbb{E}} H\left(p(y \,|\, c), q(y \,|\, c)\right) - \underset{p(z)}{\mathbb{E}} H\left(p(c \,|\, z), q(c \,|\, z)\right) - \beta I(X;C). \quad (5)$$

If we treat the entropies of the concepts and the labels as constants, we obtain

$$\mathcal{L}_{\text{E-CIB}} = \underset{p(c)}{\mathbb{E}} H\left(p(y \,|\, c), q(y \,|\, c)\right) + \underset{p(z)}{\mathbb{E}} H\left(p(c \,|\, z), q(c \,|\, z)\right) - \beta\left(\rho - I(X;C)\right), \quad (6)$$

where $\rho$ is a constant. Moreover, we found empirical evidence that also supports our assumption which we discuss in Appendix C. We denote the models that use this loss as $\text{IB}_E$ since it relies on the estimator of the mutual information. We detail the estimator we used in our implementation in Appendix D.4.

It is interesting to highlight that other optimization approaches emerge out of our bound. For instance, Kawaguchi et al. (2023) proposed $\mathcal{L}_{\text{K}} = \mathbb{E}_{p(z)} H\left(p(y \,|\, z), q(y \,|\, z)\right) + \beta(\rho - I(Z;X))$. Our mutual information estimated loss (6) resembles that proposal with the corresponding conditioning changes in the labels and the concepts. In contrast, our proposal is a generalized framework that encompass a wide range of possible implementations.

**Comparison between our estimators.** Unlike $\mathcal{L}_{\text{S-CIBM}}$ (4), which simplifies the mutual information terms into cross-entropy losses, $\mathcal{L}_{\text{E-CIB}}$ retains an explicit control over $I(X;C)$. This allows for more granular control over the information flow from inputs to concepts, leading to a tighter constraint on concept leakage. As we show in the results (Table 1), this additional control translates to improved performance in both concept and class prediction accuracy, cf. Section 4.

## 4 EXPERIMENTS

We extend several CBM variants with our IB-regularizers, yielding CIBMs. The CIBMs are slight variations of the original models as they require a variational approximation in order to study and apply the proposed IB-regularizers. We train each model from scratch and compare CIBMs to their vanilla counterparts of equal capacity, measuring both class-prediction accuracy and concept leakage. Our goal is to close the accuracy gap to black-box models without sacrificing interpretability or test-time intervenability. Finally, we analyze information flows via mutual-information estimates and benchmark intervention performance.

We benchmark our approach on three datasets: CUB (Wah et al., 2011), AwA2 (Xian et al., 2019), and aPY (Farhadi et al., 2009). We present all implementation details in Appendix D. For our regularizers, we evaluate their setups and select the best hyperparameters (cf. Section 4.6). In the following experiments, we use the same hyperparameters and setup for our regularizers for fair comparisons.

### 4.1 PERFORMANCE ACROSS ALL DATASETS

We present the evaluation results on three datasets in Table 1. Our "black-box model" serves as a gold-standard: it represents the highest possible class accuracy that can be achieved by a CBM when it is trained only to predict class labels (i.e., without providing any explanations). Because these models are seminal, we compare against hard (H) and soft (S) CBMs trained jointly (J) or independently (I) (Havasi et al., 2022). We discuss their reproducibility and training stability in Appendices D.2 and D.3, respectively. We also compare against more recent CBM variants such as ProbCBMs (Kim et al., 2023), intervention-aware CEM (IntCEM) (Espinosa Zarlenga et al., 2023b), and AR-CBM (Havasi et al., 2022).

Our main objective is to show that the proposed regularizers ($\text{IB}_B$ and $\text{IB}_E$) *maintain or improve* target-prediction accuracy relative to their original counterparts, while simultaneously enhancing concept-prediction accuracy and reducing concept leakage. The latter is crucial for guaranteeing the explainability of the results.

On the *CUB* dataset, both $\text{IB}_B$ and $\text{IB}_E$ improve class-prediction accuracy over all baselines and always yield better class performance. These gains are accompanied by higher (or, in the worst case, comparable) concept

**Table 1:** We report accuracy results (over 5 runs) of our proposed regularizers, $\text{IB}_B$ and $\text{IB}_E$, applied to different CBMs. Black-box is a gold standard for class prediction that offers no explainability over the concepts.

| Method | Concept | Class |
|---|---|---|
| **CUB** | | |
| Black-box | – | 0.919±0.002 |
| CBM (HJ) | 0.956±0.001 | 0.650±0.002 |
| CBM (HJ) + $\text{IB}_B$ | 0.951±0.005 | 0.654±0.000 |
| CBM (HJ) + $\text{IB}_E$ | 0.951±0.000 | 0.655±0.000 |
| CBM (HI) | 0.956±0.001 | 0.644±0.001 |
| CBM (HI) + $\text{IB}_B$ | 0.954±0.000 | 0.689±0.000 |
| CBM (HI) + $\text{IB}_E$ | 0.952±0.000 | 0.691±0.000 |
| CBM (SJ) | 0.956±0.001 | 0.708±0.006 |
| CBM (SJ) + $\text{IB}_B$ | 0.961±0.005 | 0.727±0.000 |
| CBM (SJ) + $\text{IB}_E$ | 0.958±0.005 | 0.726±0.000 |
| CBM (SI) | 0.956±0.001 | 0.639±0.001 |
| CBM (SI) + $\text{IB}_B$ | 0.950±0.002 | 0.643±0.000 |
| CBM (SI) + $\text{IB}_E$ | 0.958±0.000 | 0.646±0.000 |
| ProbCBM | 0.956±0.001 | 0.718±0.005 |
| ProbCBM + $\text{IB}_B$ | 0.960±0.000 | 0.741±0.006 |
| ProbCBM + $\text{IB}_E$ | 0.962±0.000 | 0.734±0.002 |
| IntCEM | 0.954±0.001 | 0.759±0.002 |
| IntCEM + $\text{IB}_B$ | 0.957±0.000 | 0.778±0.000 |
| IntCEM + $\text{IB}_E$ | 0.959±0.000 | 0.773±0.000 |
| AR-CBM | 0.956±0.002 | 0.761±0.010 |
| AR-CBM + $\text{IB}_B$ | 0.961±0.000 | 0.777±0.008 |
| AR-CBM + $\text{IB}_E$ | 0.950±0.000 | 0.774±0.006 |
| **AwA2** | | |
| Black-box | – | 0.893±0.000 |
| CBM (HJ) | 0.979±0.000 | 0.853±0.002 |
| CBM (HJ) + $\text{IB}_B$ | 0.968±0.000 | 0.845±0.000 |
| CBM (HJ) + $\text{IB}_E$ | 0.970±0.000 | 0.856±0.001 |
| CBM (HI) | 0.979±0.000 | 0.836±0.001 |
| CBM (HI) + $\text{IB}_B$ | 0.966±0.000 | 0.823±0.000 |
| CBM (HI) + $\text{IB}_E$ | 0.976±0.000 | 0.834±0.000 |
| CBM (SJ) | 0.979±0.001 | 0.876±0.001 |
| CBM (SJ) + $\text{IB}_B$ | 0.978±0.003 | 0.878±0.000 |
| CBM (SJ) + $\text{IB}_E$ | 0.980±0.000 | 0.874±0.002 |
| CBM (SI) | 0.979±0.000 | 0.852±0.004 |
| CBM (SI) + $\text{IB}_B$ | 0.967±0.000 | 0.847±0.005 |
| CBM (SI) + $\text{IB}_E$ | 0.969±0.000 | 0.853±0.003 |
| ProbCBM | 0.979±0.000 | 0.880±0.003 |
| ProbCBM + $\text{IB}_B$ | 0.971±0.001 | 0.883±0.000 |
| ProbCBM + $\text{IB}_E$ | 0.973±0.000 | 0.883±0.000 |
| IntCEM | 0.979±0.000 | 0.884±0.002 |
| IntCEM + $\text{IB}_B$ | 0.977±0.000 | 0.884±0.000 |
| IntCEM + $\text{IB}_E$ | 0.982±0.000 | 0.886±0.003 |
| AR-CBM | 0.979±0.001 | 0.884±0.006 |
| AR-CBM + $\text{IB}_B$ | 0.977±0.004 | 0.876±0.006 |
| AR-CBM + $\text{IB}_E$ | 0.978±0.004 | 0.879±0.000 |
| **aPY** | | |
| Black-box | – | 0.866±0.003 |
| CBM (SJ) | 0.967±0.000 | 0.797±0.007 |
| CBM (SJ) + $\text{IB}_B$ | 0.966±0.000 | 0.850±0.000 |
| CBM (SJ) + $\text{IB}_E$ | 0.968±0.000 | 0.860±0.006 |
| CBM (SI) | 0.967±0.000 | 0.792±0.007 |
| CBM (SI) + $\text{IB}_B$ | 0.966±0.000 | 0.845±0.012 |
| CBM (SI) + $\text{IB}_E$ | 0.957±0.000 | 0.848±0.000 |
| ProbCBM | 0.967±0.000 | 0.863±0.007 |
| ProbCBM + $\text{IB}_B$ | 0.960±0.000 | 0.873±0.005 |
| ProbCBM + $\text{IB}_E$ | 0.958±0.002 | 0.868±0.000 |
| IntCEM | 0.967±0.000 | 0.869±0.004 |
| IntCEM + $\text{IB}_B$ | 0.966±0.000 | 0.868±0.000 |
| IntCEM + $\text{IB}_E$ | 0.971±0.000 | 0.877±0.002 |
| AR-CBM | 0.967±0.000 | 0.873±0.004 |
| AR-CBM + $\text{IB}_B$ | 0.963±0.003 | 0.871±0.006 |
| AR-CBM + $\text{IB}_E$ | 0.968±0.000 | 0.879±0.007 |

accuracy, thereby, achieving the fundamental goal of our approach: to boost performance *and* interpretability at the same time. The CBM families (Havasi et al., 2022) were trained for 100 epochs, in line with current training protocols (Espinosa Zarlenga et al., 2022; Kim et al., 2023); reproducibility details are provided in Appendix D.2.

For the *AwA2* dataset, the increase in class accuracy is modest but still comparable to that of the original methods, and concept prediction remains on par with the unregularized models. We attribute this to the relative simplicity of the dataset, which leaves little room for substantial improvement.

In the more diverse, real-world *aPY* dataset, our regularizers significantly outperform the baseline CBMs in class accuracy. Remarkably, they even surpass the black-box model while delivering interpretability comparable to the original CBMs—an essential property for real-world applications where explanations are required.

The overall rise in both class and concept accuracy relative to existing methods highlights the benefits of our mutual-information regularization. By curbing concept leakage, the regularizers ensure that concepts are both informative and tightly coupled to the final prediction (see Section 4.2 for details). This empirical finding aligns with our theoretical framework, which argues that controlling the information flow between inputs and concepts via the Information Bottleneck produces more interpretable and meaningfully useful concepts without sacrificing performance (see Section 4.5 for a deeper discussion).

## 4.2 CONCEPT LEAKAGE

Concept leakage occurs when spurious or task-irrelevant information contaminates concept activations, thereby eroding both interpretability and the effectiveness of test-time interventions (Mahinpei et al., 2021; Margeloiu et al., 2021). Espinosa Zarlenga et al. (2023a) introduced two complementary metrics for quantifying this phenomenon: the Oracle Impurity Score (OIS), which measures impurity localized within individual concepts, and the Niche Impurity Score (NIS), which captures impurity distributed across the whole set of learned concepts.

We evaluate OIS and NIS under three different settings: (i) *Complete concept set*: all concepts are retained. (ii) *Selective dropout*: we remove the *most predictive* half of the concepts (i.e., the half with the highest class-prediction importance). This scenario has a unique configuration, so we do not report a standard deviation. (iii) *Random dropout*: we randomly discard half of the concepts, serving as a control condition.

We chose dropout experiments because omitting relevant concepts can dramatically increase concept leakage (Havasi et al., 2022). Tables 2, G.1, and G.2 present the results. In every scenario our IB-regularizers ($IB_B$ and $IB_E$) reduce leakage substantially, achieving the lowest OIS and NIS scores even when a large fraction of concepts is removed. These findings confirm that imposing an information bottleneck on the concepts effectively curtails spurious encoding and mitigates concept leakage.

## 4.3 INTERVENTIONS

A key advantage of Concept Bottleneck Models (CBMs) is the possibility of *test-time interventions*: a user can overwrite predicted concepts with their ground-truth values and, thereby, improve the final decision. To evaluate this capability we simulate interventions by replacing the model's predicted concepts with the corresponding ground-truth concepts at test time.

Following prior work (Kim et al., 2023; Koh et al., 2020), we intervene on groups of concepts rather than on individual concepts. Grouping lets us measure how cumulative corrections affect class-prediction performance and mirrors the practical scenario where a user corrects several related concepts at once. To intervene, concept groups are sampled uniformly at random; the procedure is repeated five times and the results are averaged to reduce variance. For each number of intervened groups we compute the classification accuracy and plot it against the number of groups. The curve visualizes how quickly performance improves as more concept groups are corrected.

Figure 3 shows that CBMs regularized with our information-bottleneck penalties ($IB_B$ and $IB_E$) exhibit a *monotonic rise* in accuracy as each additional concept group is intervened. This smooth ascent demonstrates that the models truly leverage accurate concept information and suffer minimal

**Table 2:** Concept leakage evaluation (lower is better) on CUB.

| Model | Complete CS | | Selective Drop-out CS | | Random Drop-out CS | |
|---|---|---|---|---|---|---|
| | OIS | NIS | OIS | NIS | OIS | NIS |
| CBM (HJ) | $4.31 \pm 0.57$ | $65.03 \pm 1.84$ | 15.71 | 77.02 | $13.10 \pm 1.20$ | $73.09 \pm 2.12$ |
| CBM (HJ) +$IB_B$ | $2.59 \pm 0.65$ | $60.02 \pm 1.28$ | 12.67 | 70.86 | $10.32 \pm 0.57$ | $70.38 \pm 2.12$ |
| CBM (HJ) +$IB_E$ | $2.56 \pm 0.47$ | $59.52 \pm 1.91$ | 12.47 | 71.09 | $10.36 \pm 0.80$ | $69.95 \pm 1.35$ |
| CBM (HI) | $3.77 \pm 0.58$ | $65.15 \pm 2.03$ | 14.27 | 74.16 | $12.31 \pm 1.04$ | $71.78 \pm 1.86$ |
| CBM (HI) +$IB_B$ | $3.58 \pm 0.76$ | $64.95 \pm 1.59$ | 14.01 | 73.16 | $12.63 \pm 1.31$ | $70.89 \pm 2.04$ |
| CBM (HI) +$IB_E$ | $3.69 \pm 0.71$ | $64.89 \pm 1.51$ | 14.00 | 73.13 | $12.60 \pm 1.29$ | $70.93 \pm 2.09$ |
| CBM (SJ) | $4.69 \pm 0.43$ | $66.25 \pm 2.31$ | 16.29 | 78.39 | $12.97 \pm 0.78$ | $74.19 \pm 1.04$ |
| CBM (SJ) +$IB_B$ | $2.01 \pm 0.07$ | $62.32 \pm 1.95$ | 13.26 | 72.80 | $10.95 \pm 1.49$ | $71.94 \pm 0.98$ |
| CBM (SJ) +$IB_E$ | $2.00 \pm 0.09$ | $62.31 \pm 2.08$ | 13.23 | 72.85 | $11.03 \pm 1.38$ | $72.03 \pm 1.07$ |
| IntCEM | $8.74 \pm 0.30$ | $75.41 \pm 3.83$ | 20.85 | 80.19 | $18.31 \pm 0.09$ | $76.56 \pm 2.00$ |
| IntCEM +$IB_B$ | $6.15 \pm 0.20$ | $69.96 \pm 2.22$ | 17.22 | 76.67 | $14.07 \pm 0.43$ | $72.60 \pm 2.40$ |
| IntCEM +$IB_E$ | $6.07 \pm 0.23$ | $71.28 \pm 1.87$ | 18.04 | 75.75 | $13.98 \pm 0.29$ | $73.89 \pm 1.99$ |
| AR-CBM | $3.90 \pm 0.27$ | $62.30 \pm 1.52$ | 14.16 | 63.40 | $12.58 \pm 0.86$ | $60.86 \pm 1.32$ |
| AR-CBM + $IB_B$ | $2.84 \pm 0.26$ | $59.84 \pm 1.48$ | 10.96 | 59.66 | $10.17 \pm 0.48$ | $56.31 \pm 1.33$ |
| AR-CBM + $IB_E$ | $2.72 \pm 0.16$ | $59.96 \pm 0.98$ | 11.09 | 60.77 | $11.01 \pm 1.05$ | $57.23 \pm 0.79$ |
| ProbCBM | $4.30 \pm 0.10$ | $64.22 \pm 1.04$ | 16.01 | 76.92 | $13.81 \pm 0.21$ | $75.01 \pm 0.86$ |
| ProbCBM + $IB_B$ | $2.43 \pm 0.38$ | $60.29 \pm 2.02$ | 13.04 | 72.76 | $10.29 \pm 0.50$ | $70.86 \pm 1.87$ |
| ProbCBM + $IB_E$ | $2.57 \pm 0.52$ | $59.81 \pm 1.74$ | 13.05 | 73.08 | $11.15 \pm 0.62$ | $71.13 \pm 2.09$ |

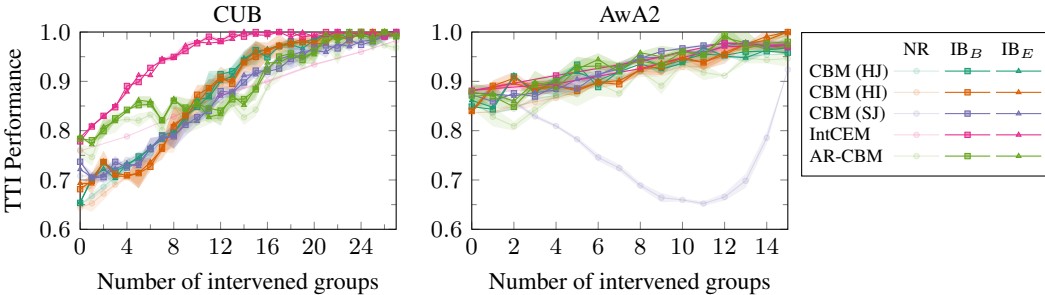

**Figure 3:** Change in target prediction accuracy after intervening on concept groups following the random strategy as described in Section 4.3. (TTI stands for Test-Time Interventions, and NR for non-regularized.) We show expanded plots, with less clutter, in Fig. F.1.

leakage: every intervention yields a consistent performance boost. In contrast, soft-joint CBMs display pronounced dips in the middle of the sequence, which we attribute to their more leaky representations that become unstable under random group corrections. Hard CBMs, which use binary concept slots, eventually reach high accuracy when a large fraction of concepts is intervened (their low leakage helps), but they start far below the CIBMs and improve much more slowly when only a few concepts are correctedespecially on coarser datasets such as AwA2.

Overall, our IB-regularised models combine low-leakage encodings with adaptive flexibility, producing steady gains and outperforming every CBM variant both in the intervention curves and in the overall accuracy reported in Table 1. Full experimental details are provided in Appendix I. Interestingly, the recent methods IntCEM and AR-CBM benefit the most from our regularization, showing a marked improvement on both datasets.

## 4.4 CONCEPT SET GOODNESS MEASURE

In CBMs, the quality of the concept set is crucial for accurate downstream task predictions. However, there is a lack of effective metrics to reliably assess concept set goodness. Existing metrics, such as the Concept Alignment Score, proposed by Espinosa Zarlenga et al. (2022), evaluate whether the model has captured meaningful concept representations but do not explicitly measure how well these concepts improve downstream task performance during interventions. Moreover, this metric is tuned for CEM and do not extend beyond it.

**Table 3:** Change in interventions performance with concept set corruption for CBM (SJ) and its regularized versions with our proposed methods. We show the disaggregated plots in Fig. F.2.

| CUB | AUC | | | NAUC | | |
|---|---|---|---|---|---|---|
| Corrupt | CBM | +IB$_B$ | +IB$_E$ | CBM | +IB$_B$ | +IB$_E$ |
| 0 | 54.374 | 65.5886 | 64.6338 | 0.001260 | 0.0691 | 0.0035 |
| 4 | 53.135 | 64.4933 | 63.4168 | 0.001198 | 0.0967 | 0.0269 |
| 8 | 51.291 | 53.1240 | 60.1120 | 0.001166 | 0.0758 | 0.0608 |
| 16 | 50.694 | 60.2241 | 59.4141 | 0.001068 | 0.0399 | 0.0451 |
| 32 | 46.101 | 52.8855 | 51.1966 | 0.000863 | 0.0045 | 0.0432 |
| 64 | 32.069 | 30.5203 | 29.2557 | 0.000339 | 0.0985 | 0.0513 |

| AwA2 | AUC | | | NAUC | | |
|---|---|---|---|---|---|---|
| Corrupt | CBM | +IB$_B$ | +IB$_E$ | CBM | +IB$_B$ | +IB$_E$ |
| No | 84.753 | 91.5144 | 92.1625 | 0.002808 | 0.0030 | 0.0115 |
| Yes | 83.985 | 90.5738 | 90.8121 | 0.004484 | 0.0023 | 0.0218 |

Similar to previous methods that rely on area under the curve for the interventions (Espinosa Zarlenga et al., 2023b; Singhi et al., 2024), we measure and compare the concept quality in CIBMs using the following metrics: area under interventions curve, and the area under curve of relative improvements. Denote by $\mathcal{I}(x)$ the model's performance for $x$ concept groups used in the intervention. Then the Test-Time Interventions (TTI) accuracy is

$$\text{AUC}_{\text{TTI}} = \frac{1}{n} \sum_{i=1}^{n} \mathcal{I}(i), \tag{7}$$

and the normalized version of the TTI accuracy is

$$\text{NAUC}_{\text{TTI}} = \frac{1}{n} \sum_{i=1}^{n} \left( \mathcal{I}(i) - \mathcal{I}(i-1) \right). \tag{8}$$

The idea behind these measures is simple: if a concept set is of high quality, the task accuracy will steadily approach $100\%$ as more concept groups are intervened upon, resulting in a large area under the curve. Conversely, if the concept set is incomplete or noisy, performance gains will be limited, even with multiple interventions, which can indicate concept leakage.

The latter expression (8) could be simplified to just scaled difference between a model with full concept set used for interventions and performance of a model with no interventions, however, the meaning it has is how much does the performance change per one group added to the interventions pool. To test this, we generate corrupted concept sets by replacing selected concepts with noisy ones. Importantly, we maintain the original groupings of concepts.

Table 3 shows the results of our metrics. We also show the commonly reported disaggregated plots in Fig. F.2. The number in the "corrupt" column denotes the number of concepts replaced with random ones for CUB, and for AwA2 "No" denotes a clear concept set and "Yes" denotes a concept set with one concept changed to corrupt. As expected, performance drops with corrupt concepts, since they contain no useful information for the target task. One consequence of our training is that if one has two concept annotations for some dataset, then it is possible to use CIBMs performance to determine which concept set is better.

Our results demonstrate that regularizing with IB$_E$ is more sensitive to concept quality compared to vanilla CBM, making it a better indicator of concept set reliability. Negative values in normalized intervention AUC indicate possible concept leakage.

## 4.5 INFORMATION PLANE DYNAMICS IN CBMS AND CIBMS

To further evaluate the proposed regularizers, we examined the information plane dynamics of CBM, IntCEM, and AR-CBM, as shown in Fig. H.1. In general, we expected to observe higher mutual information between the concepts and the labels, $I(C;Y)$, and between the latents and the concepts, $I(Z;C)$, while expecting lower mutual information between the data and the concepts, $I(X;C)$,

and between the data and the latents, $I(X;Z)$. We clearly observed this behavior when applying our $IB_E$ to IntCEM, and to a lesser degree with $IB_B$. This pattern was also evident in AR-CBMs, although with more noise. However, in certain cases, this pattern deviated. More specifically, we found that CBMs exhibit greater compression with respect to the data compared to their regularized counterparts. Nevertheless, our CIBMs demonstrate greater expressiveness due to their higher mutual information with respect to the labels, $Y$.

We think that vanilla CBMs "over-compress" their internal representations—shrinking $I(X;C)$ and $I(X;Z)$ so aggressively that they discard useful, task-relevant features. This indiscriminate bottleneck explains their lower end-to-end accuracy (Table 1) and higher concept leakage (Table 2). By contrast, our CIBMs apply a *structured* Information Bottleneck: they retain all the signal that drives $Y$—higher $I(C;Y)$—while shedding only the noise—lower $I(X;C)$—, which both boosts predictive performance and cuts leakage. In other words, achieving **expressiveness first** (then **selective compression**) yields representations that are both robust and interpretable. Appendix H presents detailed information-plane trajectories, and our findings echo recent theory on IB in deep nets, which warns against blind compression in favor of task-guided pruning (Kawaguchi et al., 2023).

Overall, we have found that pursuing compression alone is not the solution for obtaining more robust representations. Instead, we see that achieving more expressive representations (i.e., higher mutual information with respect to the labels) followed by compression (i.e., lower mutual information with respect to the data) helps reduce the gaps in predictive tasks (see Table 1) as well as in leakage (see Table 2). However, due to the requirements for expressiveness, the CIBMs do not compress as much, since they must retain some useful information. Our findings align with recent theoretical insights on the Information Bottleneck principle in deep learning (Kawaguchi et al., 2023), which emphasize that indiscriminately minimizing the mutual information between the data and the latent representations, $I(X;Z)$, does not guarantee expressive or generalizable representations. Effective models must selectively compress task-irrelevant information while retaining essential features for decision-making.

## 4.6 Evaluation of our Regularizers' Hyperparameters

We evaluated the hyperparameters of our proposed regularizers on a CBM (SJ) to select the values that we used for all other experiments. We evaluated our regularizers in a single model to find the best setup due to computational constraints. We compare the performance of $IB_B$ and $IB_E$ on concept and class prediction accuracy for the CUB dataset (using $\beta = 0.5$) and report the results in Table D.3. As shown, $IB_E$, which retains an explicit mutual information term $I(X;C)$, outperforms $IB_B$ when trained in a fair setup (vanilla) in both metrics. We found that the lack of performance of the vanilla $IB_B$ regularizer comes from instabilities during training in the latent representations encoder $p(z\,|\,x)$. We hypothesize that the gradient from the $H(C)$ in the loss (4) damages the feature encoder $p(z\,|\,x)$ since the entropy is computed w.r.t. the generative concepts $p(c)$ instead of the variational approximated ones $q(c)$. To alleviate this problem, we experimented gradient clipping as well as stopping the gradient from $H(C)$ into the encoder. We found that the latter performs on par with $IB_E$. In the experiments, we use $IB_B$ with stop gradient on it. Overall, $IB_E$'s more granular control over information flow limits concept leakage, results in better accuracies for concepts and labels in comparison to the baselines (cf. Table 1) without changes to its training framework. We also evaluate six different values for the $\beta$ constant that controls the mutual information between the data and the concepts. We show these results in Table D.4. Since we obtained inconclusive results, we selected $\beta = 0.5$ for our experiments.

These results supports our earlier discussion that the direct estimation of $I(X;C)$ leads to more effective use of concepts in downstream tasks without further changes to the training regime. Nevertheless, with a correctly regularized feature encoder $p(z\,|\,x)$, a simple estimation in $IB_B$ can achieve similar levels of information gain and accuracy.

## 5 Conclusion

We present *Concepts' Information Bottleneck Models* (CIBMs), a first-principled fusion of Information Bottleneck theory and Concept Bottleneck Models that both explains CBMs' failure modes and prescribes their cure. By penalizing $I(X;C)$ while preserving $I(C;Y)$, Concept Information

Bottleneck reveals why vanilla CBMs over-compress and leak spurious signals—and how a surgical, task-guided compression can retain exactly what matters. We validate CIBMs across six CBM families (hard/soft, joint/independent, ProbCBM, IntCEM, and AR-CBM) on three benchmarks (CUB, AwA2, and aPY), employing concept accuracy, class accuracy, Oracle and Niche Impurity (OIS and NIS), and intervention metrics ($\text{AUC}_{\text{TTI}}$, $\text{NAUC}_{\text{TTI}}$). The result is uniformly higher class accuracy, dramatically reduced concept leakage, and equal or better concept-prediction performance—closing much of the CBM-black-box gap. Crucially, our findings show that: (a) **simple, selective compression** can unlock robust, interpretable concept representations; and (b) that leakage undermines the *use* of concepts far more than their *detection*, explaining why near-perfect concept predictors can still yield subpar end-to-end performance.

## ACKNOWLEDGMENTS

This work was funded, in part, by RCN (the Research Council of Norway) through Visual Intelligence, Centre for Research-based Innovation (grant no. 309439), and FRIPRO (grant no. 359216). The computations were performed, in part, on resources provided by Sigma2—the National Infrastructure for High-Performance Computing and Data Storage in Norway (Project NN8104K).

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

# A    DETAILED DERIVATION OF CIB

In this section we present the detailed derivations to obtained the results described in Section 3.1.

We re-write the lower bound of the concepts' information bottleneck as

$$\mathcal{L}_{\text{CIBM}} = H(Y) + (1 - \beta)H(C) - H(Y \mid C) - H(C \mid Z) + \beta H(C \mid X) \tag{A.1}$$

to work with the entropies instead. We consider an approximation of the predictors for the labels and the concepts, $q(y \mid c)$ and $q(c \mid z)$, based on two variational distributions that will be implemented through neural networks—cf. Fig. 2. A conditional entropy, $H(A \mid B)$, can be expressed in terms of a variational family $q(a \mid b)$, as

$$H(A \mid B) = - \underset{p(a,b)}{\mathbb{E}} \left[ \log p(a \mid b) \right], \tag{A.2a}$$

$$= - \underset{\substack{p(a \mid b) \\ p(b)}}{\mathbb{E}} \left[ \log \frac{p(a \mid b)}{q(a \mid b)} + \log q(a \mid b) \right], \tag{A.2b}$$

$$= \underset{p(b)}{\mathbb{E}} \left[ -\mathrm{KL}\big(p(a \mid b) \,\big\|\, q(a \mid b)\big) + H\left(p(a \mid b), q(a \mid b)\right) \right], \tag{A.2c}$$

$$\leq \underset{p(b)}{\mathbb{E}} \left[ H\left(p(a \mid b), q(a \mid b)\right) \right], \tag{A.2d}$$

given that the $\mathrm{KL}\big(p(a \mid b) \,\big\|\, q(a \mid b)\big) \geq 0$. Consequently, we can re-write our conditional entropies as

$$H(Y \mid C) \leq \underset{p(c)}{\mathbb{E}} \left[ H\left(p(y \mid c), q(y \mid c)\right) \right], \tag{A.3}$$

$$H(C \mid Z) \leq \underset{p(z)}{\mathbb{E}} \left[ H\left(p(c \mid z), q(c \mid z)\right) \right]. \tag{A.4}$$

The conditional entropy of the concepts w.r.t. the data is bounded by

$$H(C \mid X) = H(C \mid X, Z) + I(C; Z \mid X), \tag{A.5a}$$

$$\geq H(C \mid Z), \tag{A.5b}$$

since $I(C; Z \mid X) \geq 0$.

Thus, the concept bottleneck loss (2) is lower bounded, given that we remove the KLs constraints, due to their positivity, from the conditional entropies (A.3), (A.4) and (A.5), and the labels entropy, by

$$\mathcal{L}_{\text{CIBM}} \geq (1 - \beta) H(C) - \underset{p(c)}{\mathbb{E}} H\left(p(y \mid c), q(y \mid c)\right) - (1 - \beta) \underset{p(z)}{\mathbb{E}} H\left(p(c \mid z), q(c \mid z)\right). \tag{A.6}$$

Moreover, we can re-write the entropy of the concepts as

$$H(C) = H(C \mid Z) + I(Z; C), \tag{A.7a}$$

$$\geq H(C \mid Z). \tag{A.7b}$$

Thus, the CIBM loss is then lower bounded by

$$\mathcal{L}_{\text{CIBM}} \geq (1 - \beta) \underset{p(z)}{\mathbb{E}} \left[ H\left(p(c \mid z)\right) - H\left(p(c \mid z), q(c \mid z)\right) \right] - \underset{p(c)}{\mathbb{E}} H\left(p(y \mid c), q(y \mid c)\right), \tag{A.8}$$

$$= -(1 - \beta) \underset{p(z)}{\mathbb{E}} \mathrm{KL}(p(c \mid z) \,\|\, q(c \mid z)) - \underset{p(c)}{\mathbb{E}} H\left(p(y \mid c), q(y \mid c)\right), \tag{A.9}$$

$$= -\mathcal{L}_{\text{S-CIBM}}. \tag{A.10}$$

We can then maximize this lower bound, $\mathcal{L}_{\text{S-CIBM}}$, to get closer to the concept bottleneck loss. In other words, we can maximize the concepts' information bottleneck by minimizing the cross entropies of the predictive variables, $y$ and $c$, and their corresponding ground truths and by maximizing the entropy of the concepts.

## B  GENERALIZATION BOUND DETAILS

To analyze the generalization of our concepts' information bottleneck models, we use the PAC-Bayes framework (McAllester, 1999; 2003).

Let $\mathcal{D} = \{(x_i, y_i, c_i)\}_{i=1}^{N}$ be a dataset of $N$ i.i.d. samples drawn from an unknown population distribution $\mathcal{P}$, such that $\mathcal{D} \sim \mathcal{P}$. Let $\theta$ be the parameters of the neural networks that approximate the encoders $q_\theta(\cdot)$.

**Proposition 1** (PAC-Bayes CBM). *For any prior $P(\theta)$, fixed before training, and a posterior $Q(\theta)$ learned by the optimization, with probability at least $1 - \delta$ over the drawn data $\mathcal{D} \sim \mathcal{P}$, we have that for all $\theta$*

$$R_{CBM}(Q(\theta)) \le \hat{R}_{CBM}(Q(\theta)) + \frac{1}{\sqrt{2N}} \sqrt{KL(Q(\theta) \parallel P(\theta)) + \log\left(\frac{2\sqrt{N}}{\delta}\right)}. \tag{B.1}$$

*Proof.* For a tuple of data $d = (x, y, c)$, we define the traditional CBM's loss as

$$\ell_{\text{CBM}}(\theta, d) = H(p(y \mid c), q_\theta(y \mid c)) + \lambda H(p(c \mid z), q_\theta(c \mid z)), \tag{B.2}$$

where $\lambda$ models different variants of CBMs. The true risk is then

$$R_{\text{CBM}}(Q(\theta)) = \mathop{\mathbb{E}}_{\theta \sim Q} \mathop{\mathbb{E}}_{d \sim \mathcal{P}} [\ell_{\text{CBM}}(\theta, d)]; \tag{B.3}$$

while the empirical risk is computed over the sampled data

$$\hat{R}_{\text{CBM}}(Q(\theta)) = \mathop{\mathbb{E}}_{\theta \sim Q} \frac{1}{N} \sum_{i=1}^{N} \ell_{\text{CBM}}(\theta, d_i), \tag{B.4}$$

where $d_i = (x_i, y_i, c_i) \in \mathcal{D}$.

Since $\mathcal{D} = \{(x_i, y_i, c_i)\}_{i=1}^{N}$ are i.i.d., the terms $\ell_{\text{CBM}}(\theta, d)$ are independent for a fixed $\theta$. Thus, by applying the standard PAC-Bayes theorem (McAllester, 1999; 2003), we obtain

$$R_{\text{CBM}}(Q(\theta)) \le \hat{R}_{\text{CBM}}(Q(\theta)) + \frac{1}{\sqrt{2N}} \sqrt{\text{KL}(Q(\theta) \parallel P(\theta)) + \log\left(\frac{2\sqrt{N}}{\delta}\right)}. \tag{B.5}$$

$\square$

**Theorem 1** (PAC-Bayes CIBM). *Any prior $P(\theta)$, fixed before training, and a posterior $Q(\theta)$ learned by the optimization, with probability at least $1 - \delta$ over the drawn data $\mathcal{D} \sim \mathcal{P}$, we have that for all $\theta$*

$$R_{CIBM}(Q(\theta)) \le \hat{R}_{CIBM}(Q(\theta)) + \frac{1}{\sqrt{2N}} \sqrt{KL(Q(\theta) \parallel P(\theta)) + \log\left(\frac{2\sqrt{N}}{\delta}\right)}. \tag{B.6}$$

*Proof.* For a tuple of data $d = (x, y, c)$, we define the CIBM's loss by expanding our concepts' information bottleneck (2) as

$$\ell_{\text{CIBM}}(\theta, d) = H(p(y \mid c), q_\theta(y \mid c)) + H(p(c \mid z), q_\theta(c \mid z)) + \beta I(X; C)$$
$$- \text{KL}(p(y \mid c) \parallel q_\theta(y \mid c)) - \text{KL}(p(c \mid z) \parallel q_\theta(c \mid z)) - H(Y) - H(C), \tag{B.7a}$$
$$= H(p(y \mid c), q_\theta(y \mid c)) + H(p(c \mid z), q_\theta(c \mid z)) + \beta I(X; C) + \mathcal{C}. \tag{B.7b}$$

For the purpose of the generalization bound, we consider the cross-entropies as the risk and the mutual information, while ignoring the reminder terms $\mathcal{C}$, such that we can compare it with the standard CBM in the following steps, since

$$\ell_{\text{CIBM}}(\theta, d) \le \ell_{\text{CBM}}(\theta, d) + \beta I(X; C), \tag{B.8}$$

since $\mathcal{C}$ is negative. Thus, the true risk is

$$R_{\text{CIBM}}(Q(\theta)) = \mathop{\mathbb{E}}_{\theta \sim Q} \mathop{\mathbb{E}}_{d \sim \mathcal{P}} [\ell_{\text{CIBM}}(\theta, d)]; \tag{B.9}$$

while the empirical risk is computed over the sampled data

$$\hat{R}_{\text{CIBM}}(Q(\theta)) = \mathop{\mathbb{E}}_{\theta \sim Q} \frac{1}{N} \sum_{i=1}^{N} \ell_{\text{CIBM}}(\theta, d_i), \tag{B.10}$$

where $d_i = (x_i, y_i, c_i) \in \mathcal{D}$.

Similarly to Proposition 1, by applying the standard PAC-Bayes theorem (McAllester, 1999; 2003), we obtain

$$R_{\text{CIBM}}(Q(\theta)) \le \hat{R}_{\text{CIBM}}(Q(\theta)) + \frac{1}{\sqrt{2N}} \sqrt{\text{KL}(Q(\theta) \parallel P(\theta)) + \log\left(\frac{2\sqrt{N}}{\delta}\right)}. \tag{B.11}$$

$\square$

**Theorem 2** (Generalization Advantage of CIBM). *Let the encoders $q(\cdot)$ be parameterized by $\theta \in \Theta$. Provided that $\beta$ is sufficiently small, the True Risk of the CIBM satisfies the inequality*

$$R(Q_{CIBM}) \le B_{CBM} - \Delta, \tag{B.12}$$

*where $B_{CBM}$ is the generalization bound of the traditional CBM, and the improvement gap $\Delta$ is defined as*

$$\Delta = \Delta_{KL}\left(\frac{1}{2\sqrt{2NKL(Q_{CBM} \parallel P) + \mathcal{K}}} - \beta\right), \tag{B.13}$$

*where $\Delta_{KL} = KL(Q_{CBM} \parallel P) - KL(Q_{CIBM} \parallel P)$, and $\mathcal{K} = \log(2\sqrt{N}/\delta)$.*

*Proof.* First, let's find the optimal distributions that minimize the empirical risk of each model. For the CBM, the optimal posterior is

$$Q_{\text{CBM}} = \arg\min_Q \hat{R}_{\text{CBM}}(Q), \tag{B.14}$$

while for CIBM

$$Q_{\text{CIBM}} = \arg\min_Q \hat{R}_{\text{CIBM}}(Q), \tag{B.15a}$$

$$= \arg\min_Q \left[\hat{R}_{\text{CBM}}(Q) + \beta I(X; C)\right], \tag{B.15b}$$

where the relation between the empirical risks (B.15b) is given by the relation between the CIBM and CBM losses (B.8). Achille & Soatto (2018) showed that the information of the representations is bounded by the information of the weights, that is,

$$I(X; C) \le I(\mathcal{D}; \theta) = \mathbb{E}_{\mathcal{D}} \text{KL}(Q(\theta \mid \mathcal{D}) \parallel P(\theta)). \tag{B.16}$$

Thus, our minimizer for the CIBM empirical risk is actually solving

$$Q_{\text{CIBM}} = \arg\min_Q \left[\hat{R}_{\text{CBM}}(Q) + \beta \mathbb{E}_{\mathcal{D}} \text{KL}(Q(\theta \mid \mathcal{D}) \parallel P(\theta))\right]. \tag{B.17}$$

In other words, the posterior $Q_{\text{CIBM}}$ that minimizes the empirical risk for the CIBM is already minimizing the posterior of the generalization bound, which is fundamentally different from the posterior minimizer of the CBM, $Q_{\text{CBM}}$, which only minimizes the risk.

Per the PAC-Bayes theorem, the True Risk $R(Q)$ for any posterior is bounded by its Empirical Risk $\hat{R}(Q)$ plus a complexity term $\mathcal{C}(Q) = \sqrt{(\text{KL}(Q \parallel P) + \mathcal{K})/2N}$, where $\mathcal{K} = \log(2\sqrt{N}/\delta)$. Therefore, for the CIBM, we start with the inequality

$$R_{\text{CIBM}}(Q_{\text{CIBM}}) \le \hat{R}_{\text{CIBM}}(Q_{\text{CIBM}}) + \frac{1}{\sqrt{2N}} \sqrt{\text{KL}(Q_{\text{CIBM}} \parallel P) + \mathcal{K}}. \tag{B.18}$$

Our goal is to substitute the terms on the right-hand side with those of the CBM to reveal the gap.

**Substituting the Empirical Risk.** Since $Q_{\text{CIBM}}$ is the global minimizer of the regularized objective (B.17), its objective value must be lower than or equal to that of any other distribution, including $Q_{\text{CBM}}$, such that

$$\hat{R}_{\text{CIBM}}(Q_{\text{CIBM}}) + \beta\text{KL}(Q_{\text{CIBM}} \parallel P) \le \hat{R}_{\text{CBM}}(Q_{\text{CBM}}) + \beta\text{KL}(Q_{\text{CBM}} \parallel P). \tag{B.19}$$

Let $\Delta_{\text{KL}} = \text{KL}(Q_{\text{CBM}} \parallel P) - \text{KL}(Q_{\text{CIBM}} \parallel P)$, such that $\Delta_{\text{KL}} > 0$ due to the explicit penalization in CIBM. Rearranging the inequality provides a bound on the increase in empirical risk as

$$\hat{R}_{\text{CIBM}}(Q_{\text{CIBM}}) \leq \hat{R}_{\text{CBM}}(Q_{\text{CBM}}) + \beta\Delta_{\text{KL}}. \tag{B.20}$$

We substitute this into the true risk bound (B.18)

$$R(Q_{\text{CIBM}}) \leq \hat{R}(Q_{\text{CBM}}) + \beta\Delta_{\text{KL}} + \frac{1}{\sqrt{2N}}\sqrt{\text{KL}(Q_{\text{CIBM}} \parallel P) + \mathcal{K}}. \tag{B.21}$$

**Substituting the Complexity Term.** Since $\Delta_{\text{KL}} > 0$ due to the information penalty, we can rewrite the complexity term of CIBM in terms of the CBM's complexity as

$$\frac{1}{\sqrt{2N}}\sqrt{\text{KL}(Q_{\text{CIBM}} \parallel P) + \mathcal{K}} = \frac{1}{\sqrt{2N}}\sqrt{\text{KL}(Q_{\text{CBM}} \parallel P) - \Delta_{\text{KL}} + \mathcal{K}}. \tag{B.22}$$

Let $A = \text{KL}(Q_{\text{CBM}} \parallel P) + \mathcal{K}$. Using the first-order Taylor approximation concavity bound of the square root function, $\sqrt{x-y} \leq \sqrt{x} - \frac{y}{2\sqrt{x}}$, we obtain

$$\frac{1}{\sqrt{2N}}\sqrt{A - \Delta_{\text{KL}}} \leq \frac{1}{\sqrt{2N}}\left(\sqrt{A} - \frac{\Delta_{\text{KL}}}{2\sqrt{A}}\right), \tag{B.23a}$$

$$= \mathcal{C}(Q_{\text{CBM}}) - \frac{\Delta_{\text{KL}}}{2\sqrt{2NA}}. \tag{B.23b}$$

**Deriving the Final Gap.** Now we substitute this upper bound for the complexity back into the true risk bound (B.21)

$$R(Q_{\text{CIBM}}) \leq \hat{R}(Q_{\text{CBM}}) + \mathcal{C}(Q_{\text{CBM}}) + \beta\Delta_{\text{KL}} - \frac{\Delta_{\text{KL}}}{2\sqrt{2NA}} \tag{B.24}$$

$$R(Q_{\text{CIBM}}) \leq B_{\text{CBM}} - \Delta_{\text{KL}}\left(\frac{1}{2\sqrt{2NA}} - \beta\right). \tag{B.25}$$

We define the gap term as

$$\Delta = \Delta_{\text{KL}}\left(\frac{1}{2\sqrt{2N(\text{KL}(Q_{\text{CBM}} \parallel P) + \mathcal{K})}} - \beta\right). \tag{B.26}$$

Provided that $\beta$ is sufficiently small such that $\beta < \frac{1}{2\sqrt{2NA}}$, then $\Delta > 0$. Thus, we conclude

$$R(Q_{\text{CIBM}}) \leq B_{\text{CBM}} - \Delta. \tag{B.27}$$

$\square$

## C  STUDY OF EMPIRICAL ENTROPY $H(C)$

In deriving the E-CIB bound (6), we simplified the formulation by treating the entropy of the concepts as approximately constant. On one hand, we assume that the concepts entropy was constant since it depends on the prior distribution of the concepts. On the other hand, this simplification is also empirically supported by our results (see Fig. C.1), which confirm that concept entropy indeed rapidly stabilizes early in training, exhibiting minimal fluctuation thereafter.

## D  IMPLEMENTATION DETAILS

### D.1  DETAILS ON THE MODELS

In the following we describe the experimental setup for each model's encoder $p(z \mid x)$. The IB-regularized variants use exactly the same configuration as their unregularized counterparts, except for the addition of the IB-regularizer (see the paragraph below for implementation details). The original *CBM family* (Havasi et al., 2022) employed an Inception backbone, and we followed this choice.

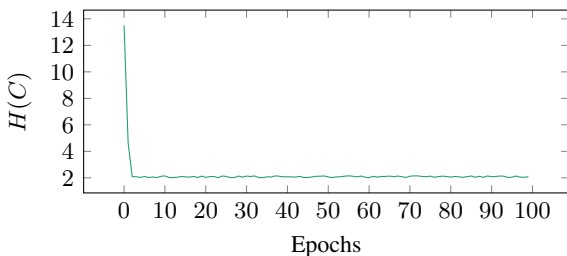

**Figure C.1:** Calculation of the empirical entropy of the concepts, $H(C)$, during training for CBM (SJ) on CUB.

Our code is based on the ProbCBM implementation (Kim et al., 2023). We trained the models for 100 epochs while the authors of CBM (Havasi et al., 2022) used 1000 epochs, but we adopt the shorter schedule that is standard in more recent works—see the sensitivity analysis in Appendix D.3). *ProbCBM* (Kim et al., 2023) uses a ResNet-18 backbone. We adopted the exact configuration and code from the original repository and trained for 100 epochs. *IntCEM* (Espinosa Zarlenga et al., 2023b) is implemented with a ResNet-34 backbone. We reused the IntCEM code and trained the model for 300 epochs. *AR-CBM* (Havasi et al., 2022) also relies on an Inception backbone. We followed the AR-CBM repository and trained for 200 epochs. All remaining hyper-parameters (learning rate, optimizer, batch size, etc.) are kept identical to the original implementations. A detailed comparison of different backbone choices and their impact on performance is provided in Appendix D.2.

To endow any existing concept-bottleneck model with our information-bottleneck regularizer we modify the layer that produces the latent representation $Z$. Specifically, we attach two lightweight heads to the model's embedding layer (the bottleneck): one head predicts a mean vector $\mu(z)$ and the other predicts a standard-deviation vector $\sigma(z)$. In a nutshell for these heads, we add on top of the model's embedding layer (the bottleneck of the model) two 1-layer MLP (i.e., our heads), for mean and standard deviation using the reparametrization trick in the variational approximation $q(c \mid z)$, each of dimensionality 112—the number of concepts left after filtration identical to one done in Koh et al.'s (2020) work. For IntCEMs, we introduce variational approximation for every concept embedding projection. We obtain concept logits as $C = \text{pred}_\mu(x) + \text{pred}_\sigma(x) \cdot \epsilon$, where $\epsilon$ is a random standard Gaussian noise. On top of concepts logits, we stack label predictor $q(y \mid c)$ (also 1-layer MLP). All activations between the layers are ReLU.

## D.2 DISCUSSION ON REPRODUCIBILITY OF PREVIOUS WORK

We emphasize that in our implementation of each baseline and their IB-regularized versions utilized the same setups (backbones and hyperparameters) as discussed in Appendix D.1. Additionally, we validated the execution of the base model and found that our results were within one standard deviation of the reported baselines in the original papers, confirming the accuracy of our baselines. Despite these sanity check values, we reported the original values, as is standard practice.

CBMs' results (Havasi et al., 2022) use two backbones: Inception and ResNet18. We retrained the CBM (SJ) and our regularizers for 100 epochs while varying the backbone and observed similar results, albeit slightly lower with ResNet18, as shown in Table D.1. For example, in the CUB dataset trained for 100 epochs, we observed a class accuracy of 0.708 with the Inception backbone compared to 0.683 with ResNet18. Considering our experiments with up to 1000 epochs (cf. Appendix D.3), we predict these results will scale similarly. Notably, the concept accuracies are comparable at 0.95, suggesting that the results reported by Kim et al. (2023) align more closely with the Inception backbone than with ResNet18.

Regarding ProbCBM (Kim et al., 2023), our results follow those using a ResNet18, aligning with the original results. In the case of IntCEM and CBM results, despite the claims in ProbCBM, we believe there are inconsistencies between the original reported results (Kim et al., 2023) and the claims regarding the backbones, as explained below.

For IntCEM, the results are remarkably similar between the ResNet18 and ResNet34 backbones, as noted by Espinosa Zarlenga et al. (2022) in Appendix A.4 (Fig. A.7) of their paper. This similarity

**Table D.1:** Effect of the backbone on the CBM (SJ) and our regularizers.

| Method | Backbone | Concept | Class |
|---|---|---|---|
| CBM | Inception | 0.956 | 0.708 |
| CBM + IB$_B$ | Inception | 0.957 | 0.726 |
| CBM + IB$_E$ | Inception | 0.957 | 0.727 |
| CBM | Resnet18 | 0.951 | 0.683 |
| CBM + IB$_B$ | Resnet18 | 0.954 | 0.701 |
| CBM + IB$_E$ | Resnet18 | 0.954 | 0.703 |

**Table D.2:** Comparison of concept and class prediction accuracies when changing the number of epochs in CUB.

| Method | Epochs | Concept | Class |
|---|---|---|---|
| CBM | 100 | 0.956 | 0.708 |
| CBM + IB$_B$ | 100 | 0.957 | 0.726 |
| CBM + IB$_E$ | 100 | 0.957 | 0.727 |
| CBM | 1000 | 0.957 | 0.795 |
| CBM + IB$_B$ | 1000 | 0.958 | 0.806 |
| CBM + IB$_E$ | 1000 | 0.958 | 0.810 |
| ProbCBM | 50 | 0.954 | 0.702 |
| ProbCBM + IB$_E$ | 50 | 0.954 | 0.710 |
| ProbCBM + IB$_B$ | 50 | 0.954 | 0.708 |
| ProbCBM | 100 | 0.956 | 0.718 |
| ProbCBM + IB$_E$ | 100 | 0.957 | 0.734 |
| ProbCBM + IB$_B$ | 100 | 0.957 | 0.732 |

might explain the scores reported in the ProbCBM (Kim et al., 2023). Additionally, there is no evidence that Kim et al. (2023) implemented and ran IntCEMs; we rely on their assertion regarding ResNet18, but there is no code-based justification. We executed IntCEM with 300 epochs, as described in the original paper, using different backbones and found accuracies of $0.757$, $0.762$, and $0.770$ for ResNet18, ResNet34, and Inception, respectively. These results support the original IntCEM paper's claim that there is no significant difference with respect to the backbone. Moreover, the results for the ResNet models fall within the standard deviation of the values reported in the ProbCBM paper. However, establishing significant results would require more runs and a proper hypothesis test, which are beyond the scope of our work.

### D.3 CBMs SENSITIVITY TO TRAINING EPOCHS

In the main experiments, we trained the CBMs for 100 epochs following a protocol similar to ProbCBMs (Kim et al., 2023). However, the original CBMs (Havasi et al., 2022) were trained for 1000 epochs. In Table D.2, we reproduced results from Havasi et al.'s (2022) work using their exact setups, including the number of training epochs. As observed, our proposed IB-regularized models obtain better concept and class predicitons overall. Interestingly, the concept predictions saturate when training for 1000 epochs, while the class predictions improve. This may be due to overfitting for class prediction. These results validate that our main experiments are valid and comparable despite using fewer epochs for the CBMs. We highlight that the other methods follow the original setup that trained for hundreds of epochs as well.

### D.4 ESTIMATORS DETAILS

**Mutual Information Estimator.** Before each gradient update, we compute cross-entropies over the current batch $B_c$, and then randomly sample batch $B'_c$ from the training dataset to estimate $I(X;C)$ on this batch.

Our mutual information estimator follows Kawaguchi et al.'s (2023) work. We rely on the fact that concepts logits have Gaussian distribution for estimation of $\log p(c \,|\, x)$. And then, we use the random samples $B'_c$ to approximate the marginal of the concepts $\log p(c)$. The mutual information $I(C;X)$ is then a Monte-Carlo estimate of $\log p(c \,|\, x) - \log p(c)$.

**Table D.3:** Accuracies of CBM (SJ) with our proposed regularizers, $\text{IB}_B$ and $\text{IB}_E$, on CUB dataset (avg. 3 runs).

| Method | Concept | Class |
|---|---|---|
| $\text{IB}_B$ (vanilla) | 0.934 | 0.608 |
|     (clip_norm = 1.0) | 0.948 | 0.663 |
|     (clip_norm = 0.1) | 0.948 | 0.650 |
|     (stop grad. from $H(C)$ into $p(z \mid x)$) | 0.958 | 0.726 |
| $\text{IB}_E$ | 0.958 | 0.727 |

**Table D.4:** Evaluation of CBM (SJ) with the proposed regularizers on three datasets with two different values of $\beta$.

| | | **CUB** | | **AwA2** | | **aPY** | |
|---|---|---|---|---|---|---|---|
| $\beta$ | **Method** | **Concept** | **Class** | **Concept** | **Class** | **Concept** | **Class** |
| 0.10 | $\text{IB}_B$ | 0.957 | 0.726 | 0.978 | 0.879 | 0.967 | 0.851 |
| | $\text{IB}_E$ | 0.958 | 0.728 | 0.979 | 0.881 | 0.967 | 0.854 |
| 0.20 | $\text{IB}_B$ | 0.957 | 0.726 | 0.980 | 0.880 | 0.967 | 0.853 |
| | $\text{IB}_E$ | 0.958 | 0.727 | 0.978 | 0.880 | 0.967 | 0.852 |
| 0.25 | $\text{IB}_B$ | 0.958 | 0.725 | 0.980 | 0.886 | 0.967 | 0.850 |
| | $\text{IB}_E$ | 0.958 | 0.727 | 0.980 | 0.885 | 0.967 | 0.858 |
| 0.50 | $\text{IB}_B$ | 0.958 | 0.725 | 0.979 | 0.885 | 0.967 | 0.856 |
| | $\text{IB}_E$ | 0.959 | 0.729 | 0.979 | 0.883 | 0.967 | 0.856 |
| 0.75 | $\text{IB}_B$ | 0.958 | 0.724 | 0.979 | 0.882 | 0.967 | 0.853 |
| | $\text{IB}_E$ | 0.958 | 0.724 | 0.980 | 0.884 | 0.966 | 0.854 |
| 0.90 | $\text{IB}_B$ | 0.957 | 0.722 | 0.980 | 0.884 | 0.967 | 0.856 |
| | $\text{IB}_E$ | 0.958 | 0.726 | 0.979 | 0.883 | 0.967 | 0.855 |

**Entropy Estimator.** Since concepts $C$ are distributed normally, we use $H(C) = \frac{D}{2}(1 + \log(2\pi)) + \frac{1}{2}\log|\Sigma|$. For simplicity (since the number of concepts $D$ is constant throughout the training and inference) we use $\hat{H}(C) = \frac{1}{2}\log|\Sigma| = \sum \log(\sigma_i)$ since $\Sigma$ is a diagonal matrix in our setup.

## D.5 TRAINING PARAMETERS

We explained the hyperparameter selection in Section 4.6. We experimented with different setups to find the best configuration. We show these results in Tables D.3 and D.4. The other training parameters for the models are as follows. We set batch size to 128 and number of samples for MI estimation to 64. For all experiments we used Adam (Kingma & Ba, 2015) optimizer with $lr = 0.003$ and $wd = 0.001$. We experimented with gradient clipping, but it led to either slow or divergent training, so we are not clipping the gradients in any of the experiments.

## D.6 DATASETS

We benchmark our approach on 3 datasets: CUB (Wah et al., 2011), AwA2 (Xian et al., 2019), and aPY (Farhadi et al., 2009). While CUB is a recognized dataset for comparing concept-based approaches (Espinosa Zarlenga et al., 2022; Kim et al., 2023; Koh et al., 2020), we add the other two datasets for additional evaluations and analysis.

**CUB.** Caltech-UCSD Birds dataset (Wah et al., 2011) is a dataset of birds images totaling in 11788 samples for 200 species. Following Koh et al.'s (2020) work, for reproducibility, we reduce instance-level concept annotations to class-level ones with majority voting. We then keep only the concept that are annotated as present in 10 classes at least after the described voting, resulting in 112 concepts instead of 312. We also employ train/val/test splits provided by Koh et al. (2020), operating with 4796 train images, 1198 val images and 5794 test images. To diversify training data, we augment the images with color jittering and horizontal flip, and resize the images to $299 \times 299$ pixels for the InceptionV3 backbone. Concept groups are obtained by common prefix clustering.

**AwA2.** Animals with attributes dataset (Xian et al., 2019) is a dataset of 37322 images of 50 animal species. For the concepts set, we follow Kim et al.'s (2023) work and keep only the 45 concepts

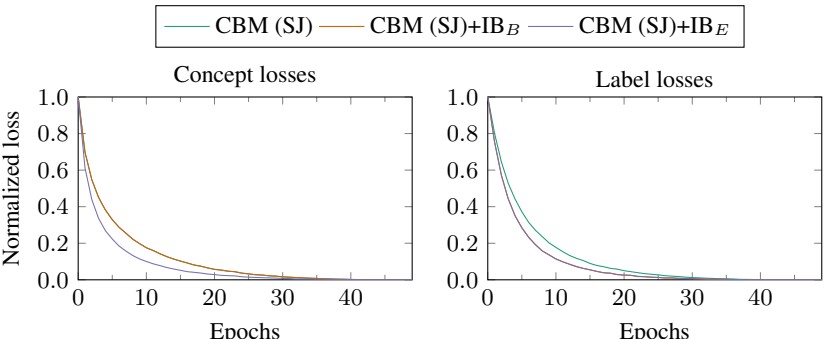

**Figure D.1:** Losses on the validation set of CUB for CBM (SJ) and its variants regularized with our proposed methods.

which could be observed on the image. We use ResNet18 embeddings provided by the dataset authors and train FCN on top of them. No additional augmentations are applied to those embeddings.

**aPY.** This is a dataset (Farhadi et al., 2009) of 32 diverse real-world classes we used for proof of concept. We split the dataset into 7362 train, 3068 validation and 4909 test samples stratified on target labels. We train FCN on top of ResNet18 embeddings of input images provided by the dataset authors (Xian et al., 2019). No additional augmentations are applied to those embeddings.

### D.7 DETAILS ON EXPERIMENTS

The image embedder backbone is only trained for CUB dataset (Wah et al., 2011), and for AwA2 (Xian et al., 2019) and aPY (Farhadi et al., 2009) we use pre-computed image embeddings. The ground truth concept labels are binary across all dataset, but concepts predictions passed to label classifier are non-binary: we are training only (and comparing only against) models using soft concepts for class prediction.

When training models with $IB_B$, we used the $\mathcal{L}_{\text{S-CIBM}}$ (4) for better performance. We backpropagate the gradients from the cross-entropies over concepts and labels through the entire network—both backbone $q(c \mid z)$ and MLPs on top of the encoder $q(y \mid c)$. For $H(C)$, however, the situation is different: gradients from this part of the loss function are propagated only through the MLPs, $q(c \mid z)$ and $q(y \mid c)$, but not the image embedder backbone $p(z \mid x)$. We found that such (partial) "freezing" of the encoder with respect to $H(C)$ constraint dramatically improves the quality of both concepts and labels prediction. While we do not have access to the ground truth probability distribution for the concepts $p(c \mid z)$, we have access to the ground truth concept labels. Our implementation uses the a supervised cross-entropy using the ground truth labels. The concepts' predictor can be seens as a multi-label task classifier. In practice, we compute $C$ logits, then, we compute binary cross-entropy (BCE) for each of these logits with binary labels. Finally, we backpropagate them through the means of BCEs.

We show the normalized loss function values on the validation set of CUB in Fig. D.1 to show the convergence of CIBMs in comparison to CBM (SJ). Note that visually the concept losses on between CBM (SJ) and its variant regularized with $IB_E$ and the label losses between CIBMs are similar, but they differ slightly.

## E  EXTENDED RESULTS ON PREDICTION

We trained the CBMs on CUB using the soft sequential (SS) versions of our proposed method to complement the different versions of vanilla CBMs. We present the results in Table E.1. As demonstrated, our regularizers perform as expected and continue to outperform the corresponding baselines. While we report these additional results for completeness, we highlight that, most CBM-based works report the soft joint version of CBM, with only the original paper presenting the soft sequential versions.

**Table E.1:** Extended results for Table 1 on CUB. Accuracy results include mean and std. over 5 runs. We report results of our proposed regularizer methods, $IB_B$ and $IB_E$, applied to different CBMs. Black-box is a gold standard for class prediction that offers no explainability over the concepts.

| Method | Concept | Class |
|---|---|---|
| Black-box | – | 0.919±0.002 |
| CBM (HJ) | 0.956±0.001 | 0.650±0.002 |
| CBM (HJ) + IB$_B$ | 0.946±0.000 | 0.653±0.004 |
| CBM (HJ) + IB$_E$ | 0.960±0.000 | 0.647±0.000 |
| CBM (HI) | 0.956±0.001 | 0.644±0.001 |
| CBM (HI) + IB$_B$ | 0.955±0.000 | 0.683±0.000 |
| CBM (HI) + IB$_E$ | 0.948±0.002 | 0.679±0.000 |
| CBM (SI) | 0.956±0.001 | 0.639±0.001 |
| CBM (SI) + IB$_B$ | 0.955±0.000 | 0.642±0.000 |
| CBM (SI) + IB$_E$ | 0.958±0.004 | 0.642±0.009 |
| CBM (SS) | 0.956±0.001 | 0.640±0.001 |
| CBM (SS) + IB$_B$ | 0.954±0.000 | 0.673±0.008 |
| CBM (SS) + IB$_E$ | 0.955±0.000 | 0.664±0.000 |
| CBM (SJ) | 0.956±0.001 | 0.708±0.006 |
| CBM (SJ) + IB$_B$ | 0.951±0.000 | 0.724±0.002 |
| CBM (SJ) + IB$_E$ | 0.957±0.000 | 0.731±0.000 |

## F    EXTENDED RESULTS ON INTERVENTIONS

In Fig. F.1, we show the plots of Fig. 3 separated and grouped by the type of method and dataset in order to better visualize the trends. We highlight that the fewer points in the results for IntCEM follows the results from Espinosa Zarlenga et al. (2022).

In Fig. F.2, we show additional results about the aggregated interventions that we dicussed in Section 4.4 and that we showed in Table 3. We plot the interventions in the traditional way by showing the intervened groups and the TTI performance for six different corruption settings.

## G    EXTENDED RESULTS ON CONCEPT LEAKAGE

### G.1    OIS AND NIS METRICS

The Oracle Impurity Score (OIS) (Espinosa Zarlenga et al., 2023a) quantifies impurities localized within individual concept representations. Given a concept encoder $g \colon X \to \hat{C} \subseteq \mathbb{R}^{d \times k}$, test samples $\Gamma_X$, and their concept annotations $\Gamma$, OIS is defined as:

$$\text{OIS}(g, \Gamma_X, \Gamma) = \frac{2\|\pi(g(\Gamma_X), \Gamma) - \pi(\Gamma, \Gamma)\|_F}{k} \tag{G.1}$$

where $\pi(\hat{\Gamma}, \Gamma)$ is a purity matrix whose entries $\pi(\hat{\Gamma}, \Gamma)_{(i,j)}$ contain the AUC-ROC score when predicting the ground truth value of concept $j$ given the $i$-th concept representation. The normalization ensures OIS ranges in $[0, 1]$, with $0$ indicating perfect alignment between the predictive capacity of learned and ground truth concepts.

The Niche Impurity Score (NIS) (Espinosa Zarlenga et al., 2023a) captures impurities distributed across multiple concept representations. Let $f \colon \mathbb{R}^D \to C$ be a classifier, and $\mathcal{D} = \{r^{(l)}, c^{(l)}\}_{l=1}^n$ be a dataset of labeled concept representations. For each concept $i$, a concept niche $N_i(\nu, \beta)$ is defined as the set of concept indices whose representations are highly entangled with concept $i$ according to a concept niche function $\nu$ and threshold $\beta$. Let the impurity set, $\mathbb{I}_i = \{1, \ldots, d\} \setminus \mathcal{N}_i(\nu, \beta)$, as the set of all indices excluded from the niche.

The Niche Impurity (NI) for concept $i$ measures how predictable this concept is from representations outside its niche:

$$\text{NI}_i(f, \nu, \beta) = \text{AUC-ROC}\left(\left\{f_{\mathbb{I}_i}(\hat{c}_{\mathbb{I}_i}^{(l)}), c_i^{(l)}\right\}\right), \tag{G.2}$$

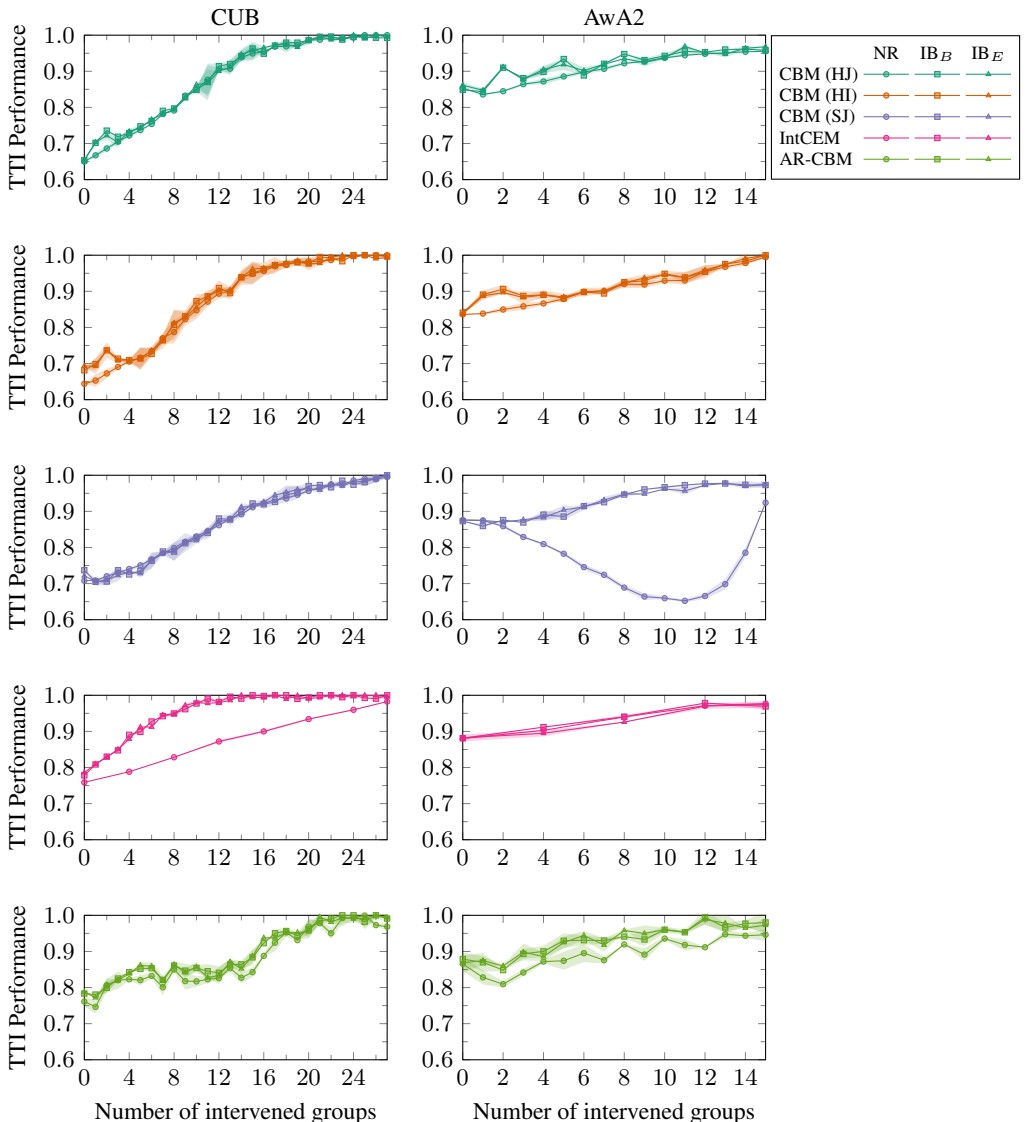

**Figure F.1:** Expanded results from Fig. 3. Change in target prediction accuracy after intervening on concept groups following the random strategy as described in Section 4.3. (TTI stands for Test-Time Interventions and NR for non-regularized.)

where $f_{\mathbb{I}_i}$ denotes the classifier restricted to operate only on the dimensions in $\mathbb{I}_i$, and $\hat{c}^{(l)}_{\mathbb{I}_i}$ denotes the sub-vector formed by selecting only the indices found in $\mathbb{I}_i$ (the irrelevant features). The overall NIS is then calculated by integrating NIs across all concepts and threshold values:

$$\text{NIS}(f, \nu) = \int_0^1 \left( \sum_{i=1}^k \frac{\text{NI}_i(f, \nu, \beta)}{k} \right) d\beta. \tag{G.3}$$

A NIS of $0.5$ indicates random performance (no impurity), while a NIS of $1$ suggests that concept information is dispersed across multiple representations. Together, these metrics effectively evaluate concept quality without making unrealistic assumptions about concept independence or representation dimensionality.

### G.2 CONCEPT SETS REDUCTION

We employed two different algorithms to cut the concepts set to half the size: selective (information-based) and random dropout. In the former, we computed $\mathbb{E}[I(Y; C_i)]$ for all concept groups on a

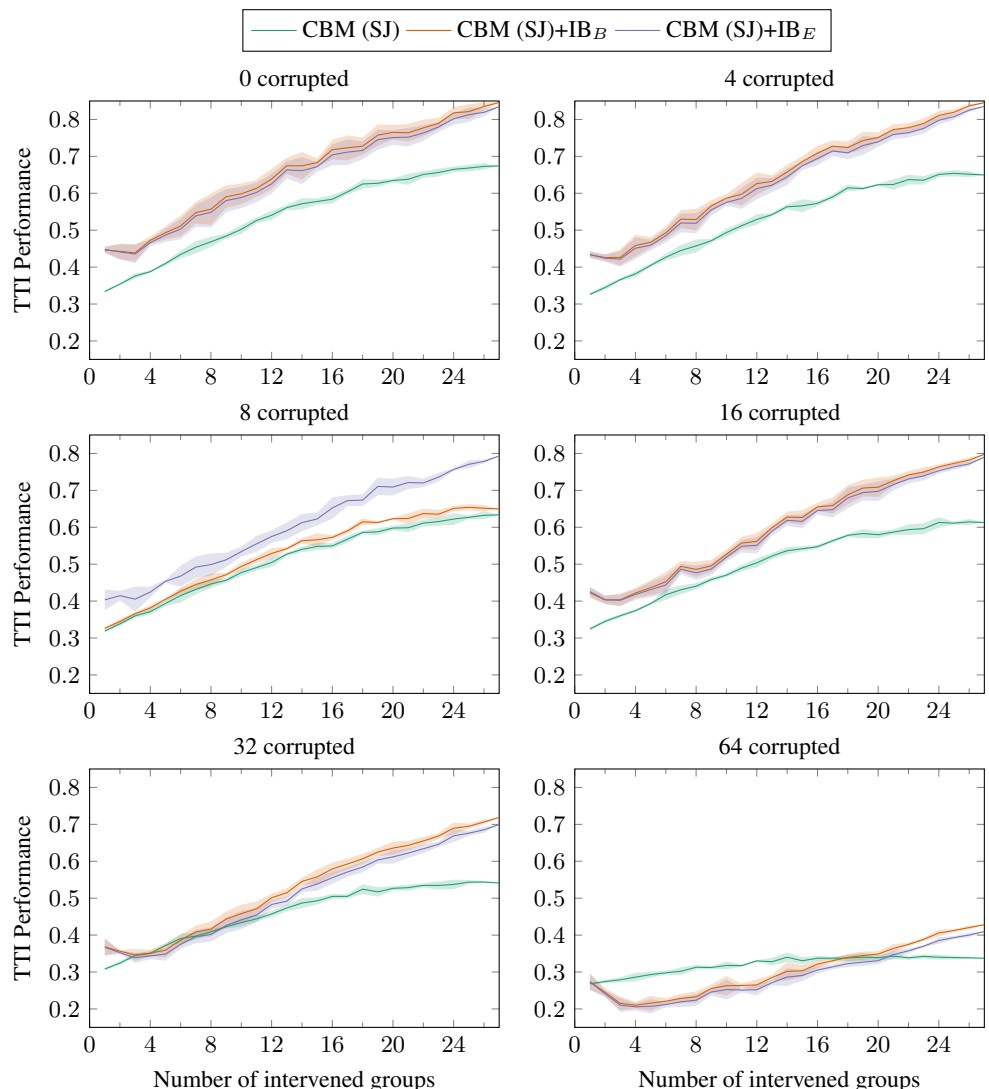

**Figure F.2:** Change in target prediction accuracy for different number of corrupted concepts. These are the expanded results of Table 3. (TTI stands for Test-Time Interventions.)

subsample of the training set. Then we dropped out the concepts groups with the highest mutual information—that is, we made the "fair" (leakage-free) learning as unprofitable and hard as possible. On the other hand, the random dropout selects half of the concepts at random and drops the rest.

## G.3 ADDITIONAL RESULTS

To complement the results of Table 2, we also performed the concept leakage evaluation on AwA2 (Table G.1 and aPY (Table G.2). These results demonstrate that our regularizers are highly effective at mitigating concept leakage, as evidenced by the consistent reduction in OIS and NIS across all three datasets (CUB, AwA2, and aPY). This improvement is robust across every tested model architecture (including various CBMs, IntCEM, and ProbCBM) and intervention strategy, indicating that these regularizers successfully force models to learn better concepts by preventing the encoding of extraneous information. Furthermore, the performance of our regularizers is similar, suggesting that both are equally reliable solutions for enforcing concept integrity without a significant difference in efficacy between them.

**Table G.1:** Concept leakage evaluation (lower is better) on AwA2.

| Model | Complete CS | | Selective Drop-out CS | | Random Drop-out CS | |
|---|---|---|---|---|---|---|
| | OIS | NIS | OIS | NIS | OIS | NIS |
| CBM (HJ) | $2.61 \pm 0.27$ | $57.98 \pm 1.15$ | 10.09 | 67.58 | $9.49 \pm 1.36$ | $70.74 \pm 1.89$ |
| CBM (HJ) + IB$_B$ | $2.29 \pm 0.23$ | $56.89 \pm 1.91$ | 9.71 | 64.37 | $9.28 \pm 0.88$ | $69.79 \pm 1.37$ |
| CBM (HI) | $2.01 \pm 0.43$ | $63.03 \pm 1.97$ | 9.27 | 64.18 | $8.71 \pm 1.03$ | $69.14 \pm 0.34$ |
| CBM (HI) + IB$_B$ | $1.78 \pm 0.34$ | $60.05 \pm 0.91$ | 8.40 | 67.23 | $8.31 \pm 1.94$ | $67.25 \pm 1.52$ |
| CBM (SJ) | $3.01 \pm 0.29$ | $64.44 \pm 1.92$ | 11.24 | 70.41 | $10.69 \pm 1.00$ | $70.68 \pm 0.57$ |
| CBM (SJ) + IB$_B$ | $2.62 \pm 0.24$ | $61.63 \pm 0.49$ | 9.52 | 68.12 | $8.94 \pm 1.52$ | $68.57 \pm 1.26$ |
| IntCEM | $6.34 \pm 0.70$ | $72.46 \pm 2.86$ | 14.74 | 74.19 | $12.16 \pm 0.24$ | $72.24 \pm 1.58$ |
| IntCEM+ IB$_B$ | $5.79 \pm 1.03$ | $68.57 \pm 2.06$ | 9.95 | 68.02 | $10.93 \pm 0.45$ | $68.94 \pm 2.04$ |
| AR-CBM | $3.90 \pm 0.27$ | $62.30 \pm 1.52$ | 14.16 | 63.40 | $12.58 \pm 0.86$ | $60.86 \pm 1.32$ |
| AR-CBM+ IB$_B$ | $2.70 \pm 0.19$ | $60.04 \pm 0.98$ | 11.05 | 60.85 | $11.03 \pm 1.02$ | $57.21 \pm 0.85$ |
| ProbCBM | $4.30 \pm 0.10$ | $64.22 \pm 1.04$ | 16.01 | 76.92 | $13.81 \pm 0.21$ | $75.01 \pm 0.86$ |
| ProbCBM+ IB$_B$ | $2.61 \pm 0.53$ | $59.86 \pm 1.83$ | 13.07 | 73.06 | $11.20 \pm 0.68$ | $71.20 \pm 2.07$ |

**Table G.2:** Concept leakage evaluation (lower is better) on aPY.

| Model | Complete CS | | Selective Drop-out CS | | Random Drop-out CS | |
|---|---|---|---|---|---|---|
| | OIS | NIS | OIS | NIS | OIS | NIS |
| CBM (HJ) | $2.66 \pm 0.19$ | $57.97 \pm 1.20$ | 10.17 | 67.54 | $9.44 \pm 1.45$ | $70.78 \pm 1.84$ |
| CBM (HJ) +IB$_B$ | $2.35 \pm 0.16$ | $56.91 \pm 1.91$ | 9.78 | 64.38 | $9.35 \pm 0.81$ | $69.70 \pm 1.34$ |
| CBM (HI) | $1.94 \pm 0.41$ | $62.97 \pm 1.91$ | 9.19 | 64.26 | $8.78 \pm 0.95$ | $69.06 \pm 0.24$ |
| CBM (HI) +IB$_B$ | $1.80 \pm 0.43$ | $60.03 \pm 0.86$ | 8.31 | 67.18 | $8.25 \pm 2.02$ | $67.30 \pm 1.59$ |
| CBM (SJ) | $2.97 \pm 0.37$ | $64.39 \pm 1.96$ | 11.23 | 70.40 | $10.72 \pm 1.01$ | $70.73 \pm 0.58$ |
| CBM (SJ) +IB$_B$ | $2.60 \pm 0.30$ | $61.55 \pm 0.55$ | 9.56 | 68.03 | $9.01 \pm 1.43$ | $68.59 \pm 1.21$ |
| IntCEM | $6.36 \pm 0.64$ | $72.48 \pm 2.89$ | 14.74 | 74.24 | $12.24 \pm 0.16$ | $72.31 \pm 1.53$ |
| IntCEM+IB$_B$ | $5.78 \pm 1.06$ | $68.66 \pm 2.09$ | 9.86 | 68.04 | $10.99 \pm 0.39$ | $68.91 \pm 1.98$ |
| AR-CBM | $3.89 \pm 0.18$ | $62.25 \pm 1.55$ | 14.17 | 63.42 | $12.62 \pm 0.87$ | $60.85 \pm 1.40$ |
| AR-CBM+ IB$_B$ | $2.72 \pm 0.20$ | $60.00 \pm 1.00$ | 11.04 | 60.92 | $11.00 \pm 1.08$ | $57.19 \pm 0.86$ |
| ProbCBM | $4.35 \pm 0.06$ | $64.19 \pm 1.13$ | 16.06 | 76.98 | $13.78 \pm 0.22$ | $74.94 \pm 0.87$ |
| ProbCBM+ IB$_B$ | $2.59 \pm 0.46$ | $59.86 \pm 1.84$ | 13.02 | 73.15 | $11.17 \pm 0.77$ | $71.23 \pm 2.00$ |

# H    INFORMATION PLANE DYNAMICS

We analyze the flow of information between inputs, $X$, latents, $Z$, concepts, $C$, and labels, $Y$, and present them in Fig. H.1. The objective of the information plane is to show the mutual information on the model variables after training. In particular, we expect to see a model with high $I(Z;C)$ and $I(C;Y)$ such that the corresponding variables are dependent on each other (maximally expressive), and simultaneously, low $I(X;C)$ and $I(X;Z)$ to show that the corresponding variables are maximally compressive. However, the compression of the variables alone, minimal $I(X;C)$ or $I(X;Z)$, does not guarantee that the important parts of the variables are being compressed and retained. Thus, we show the other experiments to complement this analysis.

IntCEM has a lower mutual information between the inputs and the latent and concept representations, $I(X;Z)$ and $I(X;C)$, than CBM (SJ). Interestingly, our regularizers reduce these mutual information while maintaining the mutual information w.r.t. the target, $I(C;Y)$ and $I(Z;C)$. However, for CBM (SJ), our methods increase the mutual information w.r.t. the data. This behavior may reflect the fact that CIBMs are optimized to retain task-relevant information while removing irrelevant or redundant information but not necessarily compressing as much—reflected in the higher $I(X;C)$ and $I(X;Z)$. Nevertheless, lower mutual information $I(X;C)$ and $I(X;Z)$ in CBMs does not necessarily indicate better compression given its lower predictive accuracy. Instead, it may reflect a failure to capture meaningful input features, resulting in noisier or less predictive concepts. Moreover, we note that the plots in Fig. H.1(f) for IB$_B$ and IB$_E$ look similar but they differ in hundredths.

For AR-CBM, the information flow is more noisy. Despite the noise, we can observe that CIBMs obtain higher mutual information w.r.t. the labels than their vanilla counterpart. While the compression w.r.t. the data is not as evident, the final mutual information w.r.t. the data is closer between the original method and its regularized versions. Nevertheless, we still observed better predictive

performance (cf. Table 1). Thus, we hypothesize that the regularizer is increasing the expressiveness of the representations with a trade-off of the compression as observed with the CBMs but not as apparent. On the other hand, the CIBMs obtain better compression-expression patterns for the latent representations, see Table H.1(d).

To demonstrate the effects of the compression patterns, we evaluate the alignment between representations and the target $I(C; Y)$ and show that CIBMs consistently outperform CBMs, and, while noisy, they show improvements over IntCEM, indicating that the retained information is both relevant and predictive—cf. Section 4.1. Additionally, CIBMs achieve better interpretability and concept quality, reinforcing that the higher mutual information is a reflection of meaningful expressiveness rather than leakage—cf. Section 4.3. This is further supported by the proposed intervention-based metrics ($\text{AUC}_{\text{TTI}}$ and $\text{NAUC}_{\text{TTI}}$) which highlight the importance of retaining task-relevant information in the concepts $C$. While CBMs exhibit lower mutual information between inputs and representations in contrast to the regularized versions, $I(X; C)$ and $I(X; Z)$, their poorer performance on these metrics, particularly under concept corruption, suggests that this lower information content stems from a failure to capture sufficient relevant features. By contrast, the higher $I(X; C)$ and $I(X; Z)$ in our CIBMs reflect the retention of meaningful pieces that contribute to better concept quality and downstream task performance. These findings demonstrate that reducing concept leakage requires selectively preserving relevant information rather than minimizing mutual information indiscriminately.

Our findings align with recent theoretical insights on the Information Bottleneck principle in deep learning (Kawaguchi et al., 2023), which emphasize that indiscriminately minimizing the mutual information between the data and the latent representations, $I(X; Z)$, does not guarantee expressive or generalizable representations. Instead, effective models must selectively compress task-irrelevant information while retaining essential features for decision-making. Our results (cf. Table 1 and Fig. H.1) support this trade-off by demonstrating that CBMs, despite lower $I(X; C)$ and $I(X; Z)$, do not necessarily achieve superior concept representations or intervention efficacy in comparison to their IB regularized counterparts. In contrast, our IB-based CBMs, which balance information retention and compression, lead to improved alignment between concepts and final predictions, reinforcing the importance of controlled, task-relevant compression rather than absolute mutual information minimization.

# I  DISCUSSION ABOUT CBMS SETUPS

Hard CBMs use hard concept representations, meaning that instead of producing a probabilistic output (as in soft concepts in soft CBM), each concept prediction is treated as a discrete binary or categorical value. These hard predictions are used as inputs to the downstream task (class prediction), making the pipeline interpretable and less expressive, thus less prone to information leakage.

When compared with soft CBMs and Soft CIBMs:

- Representation:
    - Hard CBMs: Use discrete hard values for concepts (e.g., 0 or 1 for binary concepts).
    - Soft CBMs: Use continuous values (e.g., logits or probabilities).
    - Soft CIBMs: Similar to soft CBMs but use IB to minimize irrelevant information, reducing concept leakage.
- Information Flow:
    - Hard CBMs: Compress information into discrete concept values, which prevents information leakage but risks losing useful details for downstream tasks.
    - Soft CBMs: Retain richer information but are more prone to concept leakage.
    - Soft CIBMs: Balance retaining relevant information while mitigating leakage through the IB framework.
- Interventions:
    - Hard CBMs: Explicitly rely on discrete corrections during interventions, which can have a significant impact.
    - Soft CBMs and CIBMs: Treat interventions as updates to probabilities or logits, which is more expressive, but could induce noise in concepts.

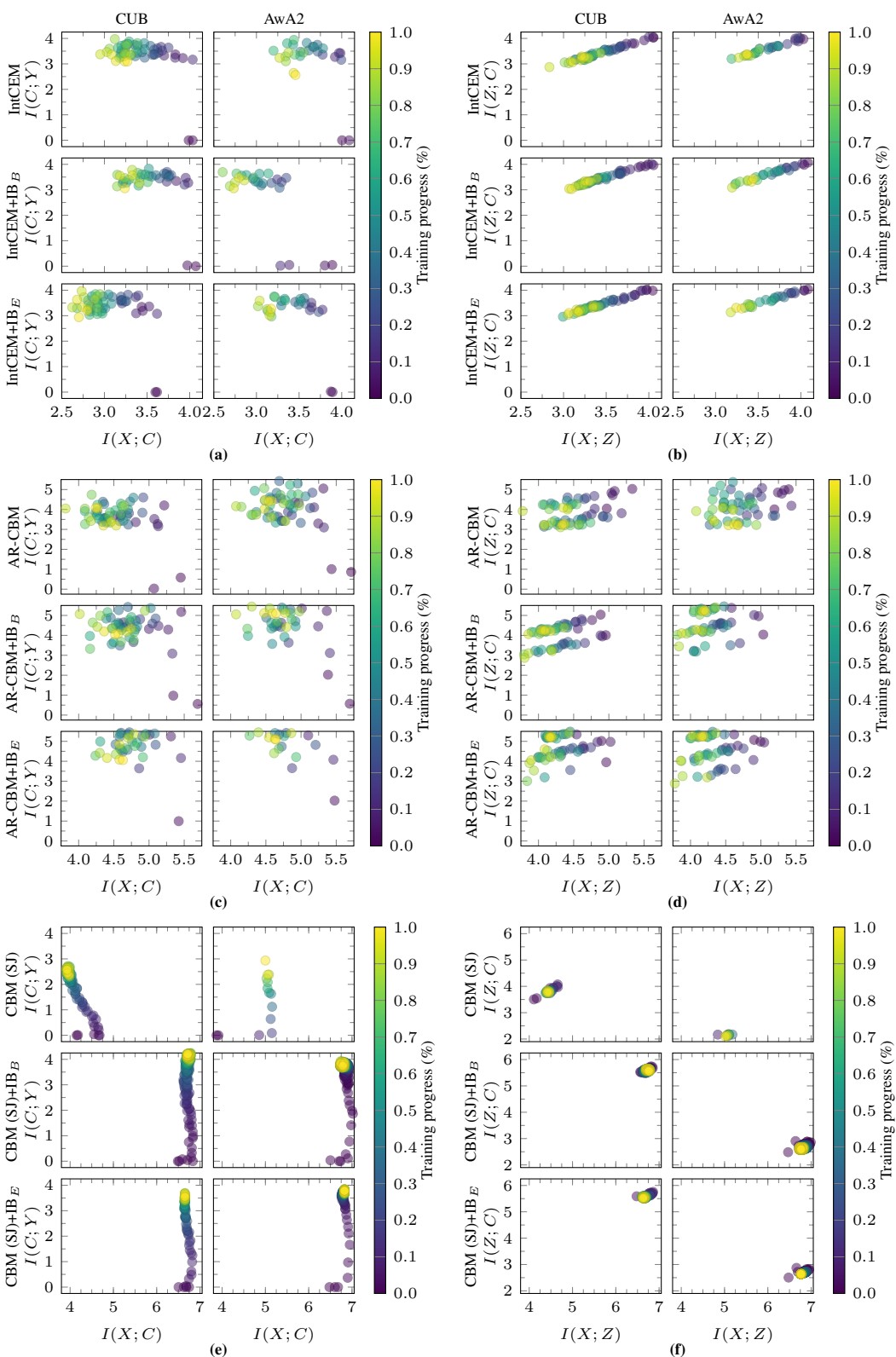

**Figure H.1:** Information plane dynamics (in nats) for (a,b) IntCEM, (c,d) AR-CBM, (e, f) CBM (SJ) and our proposed methods, $IB_B$ and $IB_E$. Warmer colors denote later steps in training. We show the information plane of (a, c, e) the variables $X$, $C$, and $Y$; and (b, d, f) the variables $X$, $Z$, and $C$.

Due to their rigidity, without enough interventions, hard CBMs cannot recover from errors or noise in the predicted concepts because the discrete pipeline does not allow for soft adjustments.

But, as more concepts are corrected, the discrete nature of hard CBMs becomes an advantage together with its independent training: ground truth, hard values fully override noisy predictions, ensuring perfect input for the downstream classifier, which was previously trained also on ground truth concepts from train set.

Soft CBMs and CIBMs, while retaining more information, still rely on probabilistic updates during interventions, which may not fully override noisy concept predictions.

Overall, CIBMs are superior because they combine the advantages of soft representations (expressiveness, better performance) with mechanisms to mitigate concept leakage (robustness, interpretability). Hard CBMs, while conceptually cleaner in avoiding leakage, fail to achieve the same level of downstream performance and adaptability, particularly in more realistic or challenging scenarios.

