# OpenReview forum: "Concepts' Information Bottleneck Models"
_ICLR.cc/2026/Conference — ICLR 2026 Poster_

### Official Review · Reviewer_L5tE · 2025-10-30

**Soundness:** 3
**Presentation:** 2
**Contribution:** 2
**Rating:** 6
**Confidence:** 4

**Summary:**

The paper introduces Concept Information Bottleneck Models, which incorporate a loss term aimed at reducing concept leakage in Concept Bottleneck Models. The objective is to maximise task-relevant information encoded in the concepts while suppressing task-irrelevant information present in the input X. The proposed regulariser is applied on top of standard CBM-like architectures. Through a series of experiments, the authors demonstrate that this additional term mitigates leakage.

**Strengths:**

The authors address a key limitation of CBMs (concept leakage) through a loss-based approach rather than architectural modifications. This design choice enhances the applicability of the proposed method, allowing it to be integrated with existing CBM variants. Notably, the models most susceptible to leakage (e.g., CEM) show the greatest improvement, further validating the effectiveness of the approach. The paper presents extensive experiments on standard benchmarks, applying the proposed regularisation to several widely used CBM-like models. Moreover, the paper is well written and easy to follow.

**Weaknesses:**

- The authors evaluate their method on datasets where the set of concepts is extensive and highly representative of both the task and the concept space. However, it would be interesting to assess how the approach performs in scenarios where some degree of leakage is necessary to solve the task. Such a setting could be simulated by systematically removing a subset of concepts (e.g., half) from the training set. This would better reflect real-world conditions, where annotating a complete set of concepts is often infeasible.
- The proposed metrics (AUC and NAUC) do not appear to represent a clear novelty. They essentially quantify what prior works have already assessed qualitatively (CBM responsiveness to concept interventions) by expressing it numerically rather than through visual plots.
- The claim of improved accuracy (Section 4.1) seems somewhat overstated. The observed gains are marginal and, arguably, accuracy should not be the primary focus of this work. It would be more appropriate to emphasise that the proposed regularisation maintains task performance while effectively reducing concept leakage, rather than suggesting accuracy improvements.

**Questions:**

- It would be interesting to explore how the proposed method behaves when the concept set is substantially reduced or incomplete (i.e., in cases where some degree of concept leakage becomes necessary to achieve good task performance). Such a setting would better approximate real-world conditions, where not all relevant concepts can be fully annotated.
- While the paper focuses on minimising information leakage from the input to the output, it does not address intra-concept leakage. Prior works [1,2] have shown that certain concepts can be predicted from others, which may undermine the independence assumption among concepts. It would be valuable to investigate whether the proposed method can also mitigate this form of leakage, thereby further improving intervention responsiveness.

---------------------------

[1] Gabriele Dominici, Pietro Barbiero, Mateo Espinosa Zarlenga, Alberto Termine, Martin Gjoreski, Giuseppe Marra, & Marc Langheinrich (2025). Causal Concept Graph Models: Beyond Causal Opacity in Deep Learning. In The Thirteenth International Conference on Learning Representations.

[2] Moritz Vandenhirtz, Sonia Laguna, Ričards Marcinkevi\vcs, & Julia E Vogt (2024). Stochastic Concept Bottleneck Models. In The Thirty-eighth Annual Conference on Neural Information Processing Systems.

---

> ### Author Response · Authors · 2025-11-25
> **First reply**
>
> We thank the reviewer for all the comments and suggestions that improved our paper.  We reply to the reviewer’s concerns below, except to the question about the necessary leakage experiments.  We are currently working on a setup and experiment as requested.  As soon as we have them, we will reply with the results.
>
> > It would be interesting to explore how the proposed method behaves when the concept set is substantially reduced or incomplete (i.e., in cases where some degree of concept leakage becomes necessary to achieve good task performance). Such a setting would better approximate real-world conditions, where not all relevant concepts can be fully annotated.
>
> We already present the reduced and noisy concept sets experiments in Table 3 for the interventions. We refer the reviewer to Section 4.3 for details about this setup and the results.
>
> > The proposed metrics (AUC and NAUC) do not appear to represent a clear novelty. They essentially quantify what prior works have already assessed qualitatively (CBM responsiveness to concept interventions) by expressing it numerically rather than through visual plots.
>
> The objective of using these metrics is to present aggregated results that are more robust to interpret as described by the reviewer.  In the paper, we do not claim novelty over them, but rather use them as a quantitative metric due to its fairness and usefulness instead of relying on qualitative visual features.
>
>
> > The claim of improved accuracy (Section 4.1) seems somewhat overstated. The observed gains are marginal and, arguably, accuracy should not be the primary focus of this work. It would be more appropriate to emphasise that the proposed regularisation maintains task performance while effectively reducing concept leakage, rather than suggesting accuracy improvements.
>
> The goal is not to improve accuracy but rather to maintain it while improving interpretability through concept robustness.  Section 4.1 is focused on accuracy due to the presentation of those results, but the understanding of the accuracy must be done in conjunction with the interpretability results where we excel.  Moreover, the class accuracy is quite saturated and thus prone to see less improvements.  On the other hand, we see improvements on most of the models we used for CUB and aPY when using one of our regularizers.
>
> Thus, we mentioned in L253 that we “maintain” the results or “improve” them.  Since, even if deemed marginal by the reviewer there is a difference.
>
> > While the paper focuses on minimising information leakage from the input to the output, it does not address intra-concept leakage. Prior works [1,2] have shown that certain concepts can be predicted from others, which may undermine the independence assumption among concepts. It would be valuable to investigate whether the proposed method can also mitigate this form of leakage, thereby further improving intervention responsiveness.
>
> We do not explicitly orthogonalise the concepts, but low OIS and NIS scores suggest disentanglement (a concept does not encode anything except for the information needed to predict itself).

---

### Official Review · Reviewer_urBx · 2025-10-30

**Soundness:** 2
**Presentation:** 3
**Contribution:** 3
**Rating:** 6
**Confidence:** 3

**Summary:**

In this work, the authors propose CIBMs which utilize an Information Bottleneck regularizer on the concept layer to reduce concept leakage for a base CBM. Evaluating their approach on different CBMs (CBM, CEM, ProbCBM, AR-CBM), the authors show strong accuracy, concept leakage, and intervention capabilities.

**Strengths:**

- Concept leakage is an important problem within CBMs, undermining faithfulness and intervention guarantees. The authors present a simple approach for mitigating leakage, utilizing an Information Bottleneck regularizer.
- Experimental details are verbose and the approach can be easily attached to prior works.

**Weaknesses:**

- Figure 3 does not really convey any information. The expanded results in the Appendix are useful but perhaps a table at different points along the x-axis would help for the main text?
- Accuracy and intervention results both show minimal improvement. Concept leakage results seem like the strongest result, but are not explored in much detail. What does the improved OIS/NIS mean downstream if the intervention performance is not improved? Is there an interpretation of these two metrics that better quantifies the improvement of CIBMs?

**Questions:**

- What dataset is Table 2 for? Why is this run on only one dataset?
- use beta=.5 for all experiments?
- "We also compare against more recent CBM variants such as ... intervention-aware CEM (Espinosa Zarlenga et al., 2022)" - is this CEM or IntCEM [1]? Why evaluate on one but not both?
- The motivation for the selected baselines in general is unclear, why is more recent work looking at intervention/leakage not included?
- Some typos throughout the main text and appendix.

[1] Espinosa Zarlenga, Mateo, et al. "Learning to receive help: Intervention-aware concept embedding models." Advances in Neural Information Processing Systems 36 (2023): 37849-37875.

---

> ### Author Response · Authors · 2025-11-25
> **First reply**
>
> We thank the reviewer for all the comments and suggestions that improved our paper.  We highlighted the changes based on their suggestions in the new version of the paper in purple.
>
> > Figure 3 does not really convey any information. The expanded results in the Appendix are useful but perhaps a table at different points along the x-axis would help for the main text?
>
> We thank the reviewer for the suggestion.  However, due to the lack of space on the main paper it is infeasible to show a table given the number of interventions, methods, and data sets. While we agree that Fig. 3 is busy, we still believe that the general trends are beneficial to the reader. Given the constraints in space, we found the trade-off of having the expanded plots in the appendix justifiable.
>
> > Accuracy and intervention results both show minimal improvement. Concept leakage results seem like the strongest result, but are not explored in much detail. What does the improved OIS/NIS mean downstream if the intervention performance is not improved? Is there an interpretation of these two metrics that better quantifies the improvement of CIBMs?
>
> While we acknowledge that the **accuracy** and **intervention results** show minimal improvements, the positive trend and the comparable performance is still a highlight. The main strength of our approach lies in the substantial improvements in concept leakage metrics (OIS/NIS), which we regard as the core contribution of our method and as direct evidence that we can reduce concept leakage while maintaining (or slightly improving) accuracy relative to the baselines.
>
> The improved OIS/NIS scores are critical because they directly quantify how well the model **isolates** concepts, which is the core goal of CBMs. While intervention performance might not be dramatically altered, lower OIS/NIS values demonstrate that the model's concepts are more **disentangled** and less reliant on spurious correlations. This guarantees that concept-level interpretation and manipulation are more **faithful** and **reliable**, thus fulfilling CBM's promise of **causal interpretability** better than previous methods.
>
>
> > What dataset is Table 2 for? Why is this run on only one dataset?
>
> CUB, as most of the benchmarking is done on it. We selected CUB given its challenging nature over the other datasets.
>
> > use beta=.5 for all experiments?
>
> Yes. We presented an ablation in the original paper in Appendix C.  In the revised paper, we refer the reviewer to Appendix D, Table D.4.
>
> > "We also compare against more recent CBM variants such as ... intervention-aware CEM (Espinosa Zarlenga et al., 2022)" - is this CEM or IntCEM [1]? Why evaluate on one but not both?
>
> We compared against IntCEM, since pure CEM is not suited for intervention as demonstrated by Zarlenga et al. (2023) themselves.
>
> > The motivation for the selected baselines in general is unclear, why is more recent work looking at intervention/leakage not included?
>
> We compared against hard CBMs and AR-CBMs that were designed to mitigate leakage.  As seen in Table 3, the proposed regularizers show consistent mitigation at leakage. As suggested by the reviewers, we are looking for more recent methods that we can compare against.  We will update the reviewer as soon as we obtain more information about these methods.

---

> > ### Comment · Reviewer_urBx · 2025-11-26
> >
> > I thank the authors for the discussion, but I still have a few main concerns.
> >
> > > we still believe that the general trends are beneficial to the reader.
> >
> > Could the authors explain what these general trends are? From my understanding the approach only helps CBM (SJ), and IntCEM on CUB. Most other models seem to be within error bounds, which is why I highlighted the busyness of the figure.
> >
> > > The improved OIS/NIS scores are critical because they directly quantify how well the model isolates concepts, which is the core goal of CBMs.
> >
> > The authors claim that better OIS/NIS scores lead to disentanglement, less reliance on spurious correlations, faithfulness, reliability, and causal interpretability - but this connection is unclear and I would warn against optimizing proxy metrics like these directly (especially when related properties such as intervention performance are unchanged). Experiments showing how CIBMs are quantifiably different from prior work on these downstream properties would significantly strengthen the work.
> >
> > > We compared against IntCEM, since pure CEM is not suited for intervention as demonstrated by Zarlenga et al. (2023) themselves.
> >
> > Seems the citation is incorrect then (2023b instead of 2022)? And why is it called CEM throughout the paper instead of IntCEM?
> >
> > > We selected CUB given its challenging nature over the other datasets.
> >
> > This is not a particularly satisfying answer. Why can the authors not also collect these results for AwA2? Is the improvement in OIS/NIS dataset-specific?

---

> > > ### Author Response · Authors · 2025-12-03
> > > **Second reply**
> > >
> > > >>  we still believe that the general trends are beneficial to the reader.
> > >
> > > > Could the authors explain what these general trends are? From my understanding the approach only helps CBM (SJ), and IntCEM on CUB. Most other models seem to be within error bounds, which is why I highlighted the busyness of the figure.
> > >
> > > We wanted to show that adding IB regularization does not damage but instead improve the model's ability to respond to concept interventions. Figure 3 shows, for every base CBM (HJ/HI/SJ, CEM, AR-CBM) and for both datasets, that $IB_B$ and $IB_E$ curves are at least as good as non regularized versions, often slightly better. On CUB you can see a clear upward separation for some base models. That's strong, visual evidence that we preserve or improve TTI performance. Moreover, given how central interventions are to CBMs, we thought keeping the figure would benefit the readers.
> > >
> > > Nevertheless, acknowledging reviewers' concern, but do not have many options on how to present it due to the lack of space.  We evaluated placing a table on the main paper instead, but the 27 interventions for CUB are too big to fit. As an alternative to alleviate the busy plot, we now added more opacity to the baselines in Fig. 3 in an attempt to declutter it.
> > >
> > >
> > > >>  The improved OIS/NIS scores are critical because they directly quantify how well the model isolates concepts, which is the core goal of CBMs.
> > >
> > > > The authors claim that better OIS/NIS scores lead to disentanglement, less reliance on spurious correlations, faithfulness, reliability, and causal interpretability - but this connection is unclear and I would warn against optimizing proxy metrics like these directly (especially when related properties such as intervention performance are unchanged). Experiments showing how CIBMs are quantifiably different from prior work on these downstream properties would significantly strengthen the work.
> > >
> > > We agree that OIS/NIS are proxy metrics. We do not claim they “prove” perfect minimal-sufficiency; rather, **consistent reductions in OIS/NIS, together with unchanged or slightly improved accuracy and TTI, strongly indicate that the model relies less on spurious, non-concept information**. For intra-concept leakage, we do not explicitly orthogonalise concepts, but the observed reductions in OIS/NIS suggest concepts become less entangled.
> > >
> > >
> > > >> We compared against IntCEM, since pure CEM is not suited for intervention as demonstrated by Zarlenga et al. (2023) themselves.
> > >
> > > > Seems the citation is incorrect then (2023b instead of 2022)? And why is it called CEM throughout the paper instead of IntCEM?
> > >
> > > We call it CEM for simplicity, but we now see that it can be misinterpreted.  We thank the reviewer for pointing out the mistake about the wrong reference.  In the revised version, we replaced CEM to IntCEM for completeness instead and cited the correct version of the paper in the experiments.
> > >
> > > >> We selected CUB given its challenging nature over the other datasets.
> > >
> > > > This is not a particularly satisfying answer. Why can the authors not also collect these results for AwA2? Is the improvement in OIS/NIS dataset-specific?
> > >
> > > We added the concept leakage results for AwA2 and aPY in the appendix Tables G.1 and G.2, respectively.  In these new results, we observed the same trends as in CUB, thus, strengthening our conclusions about the robustness of the proposed regularizers.

---

### Official Review · Reviewer_zMXB · 2025-10-31

**Soundness:** 3
**Presentation:** 3
**Contribution:** 2
**Rating:** 6
**Confidence:** 4

**Summary:**

This paper proposes to enhance the interpretability of Concept Bottleneck Models (CBMs) by applying an explicit information-bottleneck (IB) regularizer to the concept layer. It argues that traditional CBMs suffer from concept leakage (i.e., unintended information flowing into concept activations) and that minimizing I(X;C) while maximizing I(C;Y) and I(Z;C) yields cleaner and more faithful concept representations. They introduce two practical methods, integrate them into multiple CBM families and evaluate on three benchmark datasets. Experimental results suggest reductions in leakage metrics, modest improvements in classification accuracy, and more reliable concept-level interventions.

**Strengths:**

1. The paper improve target accuracy, reduce concept leakage, and enhance intervention effectiveness.
2. The information-theoretic framing is conceptually sound: targeting I(X;C) aligns well with the desired minimal-sufficient concept representation.
3. The proposed methods are modular and broadly applicable: the regularizer is applied across many CBM variants without altering the architecture radically.
4. Empirical evaluations are relatively comprehensive: multiple methods, datasets, metrics (leakage, concept accuracy, task accuracy, intervention AUC) are reported.

**Weaknesses:**

1. Stability of empirical results.
The empirical performance of different regularizers across datasets is not consistently positive.
In Table 1, certain variants of the proposed CIB regularizers improve accuracy or interpretability metrics on some datasets, but degrade or show negligible effects on others (e.g., AwA2).
This inconsistency suggests that the approach may be sensitive to dataset characteristics or model initialization, raising questions about its stability and generalization.
2. Missing theoretical justification of guaranteed improvement.
The paper introduces the modified information-bottleneck objective
$I(Z;C) + I(C;Y) - \beta I(X;C)$,
yet provides no theoretical guarantee, such as an error-bound or generalization-bound proof, showing that adding this IB term will necessarily improve the performance or faithfulness of existing CBMs.
3. Overstated claims: The paper suggests that CIBMs “close the accuracy gap to black-box models without sacrificing
interpretability,” yet in many experiments the black-box baseline still significantly outperforms the IB-regularized CBMs. The practical significance of the interpretability-performance trade-off remains underexplored.

**Questions:**

1. Could the authors provide more detailed explanation or proof of under what conditions minimizing I(X;C) while maximizing I(C;Y) ensures minimal-sufficient concept representations (i.e., no leakage)? Are there assumptions (e.g., about concept annotation completeness, model capacity, independence among concepts) required for the theory to hold?
2. In Table 1, for certain model-dataset combinations the accuracy difference between vanilla CBM and CBM+IBB/IBE is very small (e.g., <0.5%) and within one standard deviation.
3. In Eq. (5) and (7), the paper includes an entropy term H(C).
How is this quantity computed or approximated when C is continuous or high-dimensional?
Does this require batch-level estimation or additional variance reduction techniques?
4. What is the computational overhead of including IB regularization, in terms of runtime or memory?
Have the authors observed any optimization instability due to the extra MI term?

---

> ### Author Response · Authors · 2025-11-25
> **First reply (1/2)**
>
> We thank the reviewer for all the comments and suggestions that improved our paper.  We highlighted the changes based on their suggestions in the new version of the paper in orange.
>
> > Stability of empirical results. The empirical performance of different regularizers across datasets is not consistently positive. In Table 1, certain variants of the proposed CIB regularizers improve accuracy or interpretability metrics on some datasets, but degrade or show negligible effects on others (e.g., AwA2). This inconsistency suggests that the approach may be sensitive to dataset characteristics or model initialization, raising questions about its stability and generalization.
>
> Our theoretical analysis states that minimizing $I(X;C)$ and maximizing $I(C;Y)$ is most beneficial when the baseline concepts are not yet minimal-sufficient, i.e., when there is substantial concept leakage. We can see this directly in our OIS/NIS scores. For AwA2, the baseline CBMs already achieve very low impurity scores compared to CUB, indicating that the provided concept set is almost sufficient for predicting Y and that leakage is already small. In this regime, the CIBM regularizer has very little extra spurious information to strip away, so we observe minimal to no performance improvement.
>
> In contrast, on datasets like CUB where the baseline OIS/NIS values are much higher (indicating more leakage and entangled concepts), applying CIBM substantially reduces impurity and yields clear performance gains. This empirical pattern is consistent with our theoretical claim: CIBM is most effective when the initial concept representations are far from minimal-sufficient, and its impact naturally diminishes as the baseline concepts approach this ideal.
>
> In other words, the AwA2 results illustrate the conditions under which our IB objective has limited effect: when the concept set and baseline CBM are already close to a minimal-sufficient representation (low OIS/NIS), further minimizing $I(X;C)$ cannot substantially change the representation and thus has negligible impact on accuracy.
>
> > Missing theoretical justification of guaranteed improvement. The paper introduces the modified information-bottleneck objective, yet provides no theoretical guarantee, such as an error-bound or generalization-bound proof, showing that adding this IB term will necessarily improve the performance or faithfulness of existing CBMs.
>
> Following the suggestion of the reviewer, we found that the CIBM achieves a **tighter (lower) generalization bound** than the CBM (Theorem 2). This advantage is proven by showing that the complexity reduction achieved by the CIBM's information constraint (KL term) successfully outweighs the slight increase in training error ($\beta$ penalty).  We detailed the proofs and derivations in Appendix B in the revised version.  However, due to limited space in the main paper, we cannot reproduce the theorems and point the reader to the appendix for further information.
>
> > Overstated claims: The paper suggests that CIBMs “close the accuracy gap to black-box models without sacrificing interpretability,” yet in many experiments the black-box baseline still significantly outperforms the IB-regularized CBMs. The practical significance of the interpretability-performance trade-off remains underexplored.
>
> We believe this criticism stems from a misinterpretation of our core objective. Our primary goal is not to surpass the general performance ceiling set by unconstrained black-box models, but rather to maximize the performance within the constraint of providing explicit interpretability.  Unconstrained black-box models represent the uninterpretable upper bound on achievable performance. In this context, "closing the gap" refers to significantly reducing the performance difference between our constrained, interpretable CIBM and the unconstrained black-box benchmark. This highlights the inherent performance-interpretability trade-off we seek to manage, which is the central focus of our paper.

---

> > ### Author Response · Authors · 2025-11-25
> > **First reply (2/2)**
> >
> > > Could the authors provide more detailed explanation or proof of under what conditions minimizing I(X;C) while maximizing I(C;Y) ensures minimal-sufficient concept representations (i.e., no leakage)?
> >
> > The CIBM ensures minimal-sufficient concept representations (i.e., minimal leakage) by optimizing the Information Bottleneck trade-off, subject to two key conditions:
> > - Optimal Regularization: Minimal sufficiency is ensured by enforcing the compression penalty $I(X;C)$ without degrading sufficiency ($I(C;Y)$) or alignment ($I(Z;C)$). Theoretically, minimizing $I(X;C)$ bounds the information in the weights (Achille and Soato, 2018), effectively denying the model the capacity to encode "nuisance" features (leakage) from the input.
> > - Variational Validity: We assume the intractable true distributions are adequately approximated by our variational neural networks which allows the minimization of tractable cross-entropy to serve as a valid proxy for minimizing conditional entropy.  Moreover, we assume that our sampling approach approximates the expectations well-enough to compute the cross-entropies.
> >
> > Empirically, this is confirmed by our OIS/NIS scores that show our CIB regularizer provides significant gains on datasets where baselines exhibit high leakage, while showing diminishing returns on naturally "clean" baselines (e.g., AwA2), confirming that it explicitly targets and removes non-minimal information.
> >
> > > Are there assumptions (e.g., about concept annotation completeness, model capacity, independence among concepts) required for the theory to hold?
> >
> > The assumptions driving the optimization process are mainly the variational ones.  The information theoretic assumptions we used in the derivation of the gap are not required in the bounds used to train the models.
> >
> > > In Table 1, for certain model-dataset combinations the accuracy difference between vanilla CBM and CBM+IBB/IBE is very small (e.g., <0.5%) and within one standard deviation.
> >
> > While the accuracy difference is small, the results show significant improvements in interpretability (Fig. 3) and concept leakage (Table 2). Therefore, a holistic evaluation of the metrics is necessary to properly portray the model's overall strengths.
> >
> > > In Eq. (5) and (7), the paper includes an entropy term H(C). How is this quantity computed or approximated when C is continuous or high-dimensional? Does this require batch-level estimation or additional variance reduction techniques?
> >
> > In the current estimator, we assume discrete concepts and perform the entropy estimation on random batches (as detailed in Appendix D.4). We follow the entropy estimator proposed by Kawaguchi et al. (2023), which assumes a Gaussian distribution over the concepts. For the general case of continuous concepts, a similar approximation approach could be adopted, provided the parameters of the underlying Gaussian distribution are accessible.
> >
> > > What is the computational overhead of including IB regularization, in terms of runtime or memory? Have the authors observed any optimization instability due to the extra MI term?
> >
> > We did not observe instabilities on the computations of the methods.  The overhead is the computation of the estimators (i.e., the entropy and mutual information) in our methods.

---

> > > ### Comment · Reviewer_zMXB · 2025-11-26
> > >
> > > Thanks to the authors for adding tighter generalization bound and the other explanations address my concerns. I'll keep my score.

---

### Official Review · Reviewer_DEB8 · 2025-10-31

**Soundness:** 1
**Presentation:** 3
**Contribution:** 1
**Rating:** 2
**Confidence:** 4

**Summary:**

Motivated by the information bottleneck principle, the paper proposes concepts’ information bottleneck model (CIBM), which uses an information bottleneck regularizer (IB regularizer) for training concept bottleneck models (CBMs) to mitigate the problem of concept leakage while preserving prediction accuracy. Based on theoretical observations, the authors give two types of regularizers, bounded and estimator based, for training CBMs. Through experiments, it is shown that IB regularizers maintain or improve prediction performance while mitigating concept leakage. Moreover, using CIBM, the authors propose a measure to quantify the quality of concept sets.

**Strengths:**

1. The motivation for using information bottleneck principle to tackle concept leakage looks valid and interesting.
2. The experiment results suggest that IB regularizers are indeed helpful to mitigate concept leakage while preserving final prediction accuracy.

**Weaknesses:**

1. As authors frame CIBM as theoretically principled integration of IB principle to CBM, examining the validity of the estimators is important. However, theoretical analysis for deriving the bounded CIB (section 3.1, section 3.2, appendix B) seems to have fatal errors.
- The paper claims to upper bound the L_{UB-CIB}, but in eq (A.1), -\beta H(C|X) should be +\beta H(C|X), and in equations from (A.2a) to (A.4l), the minus sign should be added to all the right hand sides (definition of entropy). These errors seem to make the CIB upper bound in eq (3) or (A.5) completely wrong.
- Also, due to above observations, equation (6) seems to be wrong.
2. If the goal of the paper is to address concept leakage, comparing with previous works that address concept leakage would be a valuable addition. Also, these works do not seem to be addressed properly in the related work section.
3. At section 4.3, using only random intervention seems insufficient. The authors should include experiment that uses more effective intervention strategies explored in [1] such as UCP.

Reference

[1] Shin et al., A Closer Look at the Intervention Procedure of Concept Bottleneck Models, ICML 2023.

**Questions:**

1. Can the usage of two estimators (bounded CIB and estimator-based CIB) be theoretically justified, given that the theoretical analysis seems to be wrong?
2. How does the IB-regularizer compare against other methods that mitigate leakage?

---

> ### Author Response · Authors · 2025-11-25
> **Updated the theoretical results**
>
> We thank the reviewer for all the comments and suggestions that improved our paper.  We highlighted the changes based on their suggestions in the new version of the paper in pink.
>
> Below we reply to the reviewer’s concerns.  Regarding the additional concept leakage and intervention experiments, we are working on them and will update the replies as soon as we get them.
>
> > [...] However, theoretical analysis for deriving the bounded CIB (section 3.1, section 3.2, appendix B) seems to have fatal errors.
> > The paper claims to upper bound the L_{UB-CIB}, but in eq (A.1), -\beta H(C|X) should be +\beta H(C|X), and in equations from (A.2a) to (A.4l), the minus sign should be added to all the right hand sides (definition of entropy). These errors seem to make the CIB upper bound in eq (3) or (A.5) completely wrong.
> > Also, due to above observations, equation (6) seems to be wrong.
>
> We sincerely thank the reviewer for the identification of these errors.  After correcting the wrong sign on the $H(C|X)$, and deriving the bounds again, we obtained a final bound that still agrees with our main results.  In particular, the corrected derivation (Eq. 3 in the new version) shows that we obtain a lower bound that when maximized results in the minimization of the cross entropies of the labels and concepts (just as before) with the exception of the direction of the entropy that needs to be maximized instead. Similarly, $IB_E$ has similar results with the minimization of the cross-entropies but a maximization of the mutual information instead (Eq. 5 in the new version).  After experimenting, we observed similar results for the $IB_B$ and $IB_E$ methods as the ones we obtained previously.  We hypothesize that the entropy and mutual information regularizations help at the beginning of the learning process and quickly stabilize leaving the cross entropies as the major drivers.  Thus, the similar performance of our methods.
>
> We showed the detailed derivation and new bounds in Appendix A in the new version of the manuscript.
>
> > Can the usage of two estimators (bounded CIB and estimator-based CIB) be theoretically justified, given that the theoretical analysis seems to be wrong?
>
> Yes.  After reviewing the previous error and updating the derivation, the correct estimators are mainly optimizing the same cross-entropies and are stable.
>
> > How does the IB-regularizer compare against other methods that mitigate leakage?
>
> We compared against hard CBMs and AR-CBMs that were designed to mitigate leakage.  As seen in Table 3, the proposed regularizers show consistent mitigation at leakage.  Moreover, in the same results, we include other methods (e.g., ProbCBM and CEM) for a general overview of the leakage performance.

---

### Author Response · Authors · 2025-12-03
**Summary of the review process**

## Contributions

We propose an Information Bottleneck (IB)-based regularizer for Concept Bottleneck Models (CBMs). This is, to our knowledge, the **first work** that explicitly formulates CBMs through an IB objective and adds an IB-style loss to a broad set of CBM variants. Three reviewers (zMXB, urBx, L5tE) gave marginal accepts and found the idea interesting with solid experiments; one (DEB8) was negative mainly due to concerns about our derivations.

**Strengths.** All reviewers agree that **concept leakage is a real problem for CBMs** and that our method is simple, modular, and broadly applicable. Empirically, our regularizer **substantially reduces leakage metrics** (OIS/NIS) while **maintaining or slightly improving accuracy and intervention performance** across strong baselines (hard CBMs, AR-CBM, IntCEM). This "reduce leakage without losing performance" behavior is the main strength.

## Main concerns

Concerns focused on theory, stability of results, and baselines/metrics. We acknowledged the theoretical error and **provided corrected derivations that sustain our conclusions**. Our key value is reducing concept leakage (faithfulness) while maintaining accuracy, not maximizing raw performance.

1. **Theory**

- DEB8 **identified sign errors** in our entropy and mutual information derivations.

  A: We **re-derived all bounds** in Appendix A, corrected signs, and clarified assumptions. The corrected objectives still minimize the same cross-entropy terms plus an IB regularizer, and re-running experiments did not change qualitative conclusions.

- zMXB **requested theoretical guarantees** that the IB improves CBMs.

  A: We **added Theorem 2** in Appendix B, giving a **tighter generalization bound for CIBM than for standard CBMs**, reflecting reduced complexity under an information constraint.

2. **Experiments**

- zMXB noted that gains on AwA2 are modest compared to CUB.

  A: Our regularizers are most effective when baseline leakage is high (e.g., CUB). **AwA2 concepts are already "clean"** (low impurity), so **improvements are smaller**, which is expected (also noted by L5tE).

- urBx and L5tE argued that accuracy and intervention gains are marginal.

  A: The **goal is to close the gap to black boxes without sacrificing interpretability**, not to exceed them in accuracy. Accuracy is maintained or improved, and large OIS/NIS reductions show improved faithfulness. Intervention performance also improves when **few interventions** are performed, a realistic regime where our regularizers help most.

- DEB8 and urBx requested comparisons with newer leakage-mitigation methods and better intervention strategies (e.g., UCP).

  A: We compare against hard CBMs, AR-CBM, and IntCEM (clarifying CEM vs. IntCEM), which already target leakage. UCP-style methods are **relevant and complementary**, but fair integration requires new pipelines and strategies beyond this revision. We already evaluate **five strong CBM variants**, and our IB regularizer is orthogonal and could be combined with UCP in future work.

- urBx criticized the lack of leakage results beyond CUB.

  A: CUB was chosen for its difficulty, but we **added leakage results for AwA2 and aPY in Appendix G**. These **support our conclusion**, showing consistent leakage reduction across datasets and baselines.

3. **Concept leakage**

- urBx questioned the utility of better OIS/NIS if intervention performance does not clearly improve.

  A: Lower OIS/NIS indicates less reliance on spurious correlations and concepts closer to minimal-sufficient, which is crucial for trustworthy explanations and interventions.

- L5tE suggested addressing intra-concept leakage.

  A: We agree OIS/NIS are proxy metrics and do not claim they "prove" minimal-sufficiency; rather, **consistent OIS/NIS reductions with unchanged or slightly improved accuracy and TTI** indicate reduced use of non-concept information. We do not explicitly orthogonalize concepts, but the reductions suggest less entanglement.

4. **Presentation**

- urBx found Figure 3 too busy.

  A: The figure shows trends across multiple interventions. We increased baseline transparency and clarified that the requested large table would not fit in the main text; detailed plots are in the appendix.

## Conclusion

We addressed the main concerns by correcting derivations, introducing a tighter generalization bound, and clarifying that our primary contribution is reducing concept leakage (improving explanation faithfulness) while preserving accuracy. Our regularizers are especially effective in high-leakage settings (e.g., CUB) and safe in low-leakage regimes (e.g., AwA2), providing strong empirical support.

By framing CBMs within an IB approach that encourages minimal-sufficient concepts, we improve interpretability and reliability without compromising intervention capabilities. Because interpretable models must not rely on spurious correlations, this work is timely and merits acceptance.

---

### Meta-Review · Area_Chair_Miyf · 2026-01-01

**Summary:**

This paper proposes Concept Information Bottleneck Models (CIBMs), which introduce information bottleneck (IB) regularization at the concept layer of Concept Bottleneck Models (CBMs) to mitigate concept leakage while preserving task accuracy and intervention performance. Reviewers generally agree that concept leakage is an important and timely problem and that the proposed approach is modular and empirically promising. However, concerns were raised regarding the correctness and clarity of the theoretical derivations, the strength and interpretation of empirical gains (especially beyond leakage metrics), and the scope and justification of experimental baselines and datasets. Overall, I feel this is a borderline paper after the revision. I would strongly encourage the authors to take into account the reviewers’ comments in their revision.

**Reviewer Concerns:**

Concerns Addressed by the Rebuttal

- Theoretical errors in the bounded CIB derivation (Reviewer DEB8):
  The authors acknowledged sign errors in the entropy terms and provided corrected derivations and bounds in the revised appendix. They argued that the corrected bounds still support the proposed estimators and empirically validated similar behavior after correction. This directly addressed the reviewer’s core objection.

- Lack of theoretical justification / guarantees (Reviewer zMXB):
  The authors added a tighter generalization bound (Appendix B) and clarified the assumptions under which minimizing \( I(X;C) \) and maximizing \( I(C;Y) \) leads to minimal-sufficient concepts. Reviewer zMXB explicitly stated that these additions addressed their concerns.

- Dataset-dependent performance and stability (Reviewer zMXB):
  The rebuttal clarified that the method is most effective when baseline CBMs exhibit high leakage (e.g., CUB), and naturally has limited effect when leakage is already low (e.g., AwA2). This explanation was consistent with the empirical results and was accepted by the reviewer.

- Baseline clarity, naming, and missing results (Reviewer urBx):
  The authors corrected the CEM/IntCEM naming and citation issue, added concept leakage results for AwA2 and aPY in the appendix, and clarified baseline choices.

- Overstated accuracy claims (Reviewers zMXB, L5tE):
  The authors clarified that their primary goal is maintaining accuracy while improving interpretability, not outperforming black-box models, and adjusted the framing accordingly.

Concerns Still Outstanding

- Strength of downstream impact beyond leakage metrics (Reviewer urBx, L5tE):
  While OIS/NIS improvements are clear, the link between improved leakage metrics and stronger downstream properties (e.g., intervention effectiveness, causal faithfulness) remains somewhat indirect. Intervention performance improvements are often marginal, and the justification for treating OIS/NIS as sufficient proxies remains debatable.

- Incremental contribution and novelty (Reviewers DEB8, L5tE):
  Some reviewers remain unconvinced that the contribution goes substantially beyond existing IB-inspired regularization ideas, especially given the modest accuracy and intervention gains.

- Evaluation under incomplete or insufficient concept sets (Reviewer L5tE):
  Although the authors argue that reduced/noisy concept experiments already address this, a more explicit and systematic study of scenarios where leakage is necessary is still limited.

**Reviewer Scores:**

- Reviewer DEB8 (Original: 2 – Reject):
  *Estimated updated score: 4 or 6*
  The correction of theoretical errors and clarification of estimators likely alleviates the most severe concern, but the reviewer may still view the contribution as weak or incremental.

- Reviewer zMXB (Original: 6 – Marginal Accept):
  *Estimated updated score: 6*
  The reviewer explicitly stated they would keep their score after the rebuttal, indicating satisfaction with the responses but no change in overall enthusiasm.

- Reviewer urBx (Original: 6 – Marginal Accept):
  *Estimated updated score: 4–6*
  Despite additional clarifications, this reviewer continued to express skepticism about the interpretation of leakage metrics and the strength of empirical trends, suggesting a possible downward revision.

- Reviewer L5tE (Original: 6 – Marginal Accept):
  *Estimated updated score: 6*
  The rebuttal addressed framing and clarified goals, but some conceptual concerns (e.g., intra-concept leakage, realism of settings) likely remain.

---

### Decision · Program_Chairs · 2026-01-26

Accept (Poster)